# Data Taggants: Dataset Ownership Verification via Harmless Targeted Data Poisoning

**Wassim (Wes) Bouaziz** *
Meta AI, FAIR
& CMAP, École polytechnique
Paris, France
wesbz@meta.com

**Nicolas Usunier**
Work done while at
Meta AI, FAIR

**El Mahdi El Mhamdi**
CMAP, École polytechnique
Palaiseau, France

## Abstract

Dataset ownership verification, the process of determining if a dataset is used in a model's training data, is necessary for detecting unauthorized data usage and data contamination. Existing approaches, such as backdoor watermarking, rely on inducing a detectable behavior into the trained model on a part of the data distribution. However, these approaches have limitations, as they can be harmful to the model's performances or require unpractical access to the model's internals. Most importantly, previous approaches lack guarantee against false positives.
This paper introduces *data taggants*, a novel non-backdoor dataset ownership verification technique. Our method uses pairs of out-of-distribution samples and random labels as secret *keys*, and leverages clean-label targeted data poisoning to subtly alter a dataset, so that models trained on it respond to the key samples with the corresponding key labels. The keys are built as to allow for statistical certificates with black-box access only to the model.
We validate our approach through comprehensive and realistic experiments on ImageNet1k using ViT and ResNet models with state-of-the-art training recipes. Our findings demonstrate that data taggants can reliably detect models trained on the protected dataset with high confidence, without compromising validation accuracy, and show their superiority over backdoor watermarking. We demonstrate the stealthiness and robustness of our method against various defense mechanisms.

## 1 Introduction

An increasing number of machine learning models are deployed or published with limited transparency regarding their training data, raising concerns about their sources. This lack of transparency hinders the traceability of training data, which is crucial for assessing data contamination and the misuse of open datasets beyond their intended purpose.

Dataset ownership verification (DOV) approaches aim to equip dataset owners with the ability to track the usage of their data in specific trained models. Current methods perturb the dataset to mark models trained on it and the main challenge lies in crafting this dataset perturbation to balance two competing objectives. **On the one hand**, the perturbation should not significantly degrade performance, preserving the value of training on the data for authorized parties. **On the other hand**, the modification should sufficiently alter models' behaviors to enable high-confidence detection.
These objectives are complemented by the following technical requirements. **First**, the detection should be *effective* with *strong guarantees* against false positives (i.e. models wrongfully detected), making it challenging for dishonest model owners to claim false detection. **Second**, the perturbation should be *stealthy* to prevent easy removal by dishonest users. **Third**, it should be *robust* across different model architectures and training recipes, as to allow for usage in the wild, and adapt to the diversity of models and learning algorithms. **Finally**, for our method to be practical, the detection should be possible with *black-box* access to the model. Ideally it should work with top-$k$ predictions for small $k$, to be applicable to models available only through restricted APIs.

---

*wassim.bouaziz@polytechnique.edu

Previous works either did not provide strong theoretical guarantees (Maini, 2021; Li et al., 2022; Wenger et al., 2022), did not yield stealthy enough watermarks (Li et al., 2020b), did not demonstrate robustness of the watermark (Li et al., 2020a; Maini, 2021) or only partially allowed for black-box detection (Sablayrolles et al., 2020). Backdoor watermarking, predominantly studied in DOV literature, perturbs the dataset to predictably alter model predictions $f$ when a *trigger pattern* $x^{(trigger)}$ is added to a legitimate image $x$. This trigger should steer the prediction on the triggered sample away from the ground truth class $y$ and towards a target class $y^{(trigger)}$. The detection of suspicious models is done by running a statistical test to measure a difference between the benign prediction $f(x)_y$ and the triggered prediction $f(x + x^{(trigger)})_y$. This approach not only contradicts adversarial robustness (a desirable property for deep learning models), but is also harmful to the model as it introduces errors (Guo et al., 2023). More importantly, without proper characterization of a benign model's predictions, previous detection scheme lack theoretical grounding.

In this paper, we introduce a novel dataset ownership verification approach that enables black-box detection of dishonest models with rigorous theoretical guarantees. We call this method *data taggants* by analogy to taggants, physical or chemical markers added to materials to trace their usage or manipulation. We introduce a new detection approach: *keys*, an out-of-distribution (pattern, label) pair $(x^{(key)}, y^{(key)})$. When a model $f$ is trained on the protected dataset, it should predict $f(x^{(key)}) = y^{(key)}$ for every key. By targeting out-of-distribution key patterns $x^{(key)}$, for which a natural behavior is undefined, we limit the possibilities of inducing errors in the model contrary to backdoor watermarking. This behavior is induced through *gradient matching*, a clean-label targeted data poisoning technique introduced in Geiping et al. (2020). We perform experiments on ImageNet1k with vision transformers and ResNet architectures of different sizes, together with state-of-the-art training recipes (Wightman et al., 2021) including the 3-Augment data augmentation techniques (Touvron et al., 2022). Primarily designed for image classification datasets, similarly to prior works (Sablayrolles et al., 2020; Li et al., 2022), our main contributions are the following:

- We introduce a new detection approach: *keys*, out-of-distribution (pattern, label) pairs $(x^{(key)}, y^{(key)})$ on which we measure the top-$k$ accuracy, with $y^{(key)}$ randomly chosen.

- This randomness enables independence in the use of a theoretically more grounded Binomial significance tests, compared to previous work's use of pair-wise t-test.

- We demonstrate the *effectiveness* and practicality of data taggants through extensive experiments on ImageNet1k with state of the art training procedures. We show the *robustness* of data taggants when transferred on various model architectures and training recipes. We also bring evidence of the *stealthiness* of data taggants through PSNR, out-of-distribution (OOD) detection tests, and data poisoning defense approaches. All of this is achieved by modifying only 0.1% of the data and without degradation of performance.

- We introduce the use of a perceptual loss in the crafting of data poisons to enhance *stealthiness* and show it allows for visually imperceptible data taggants.

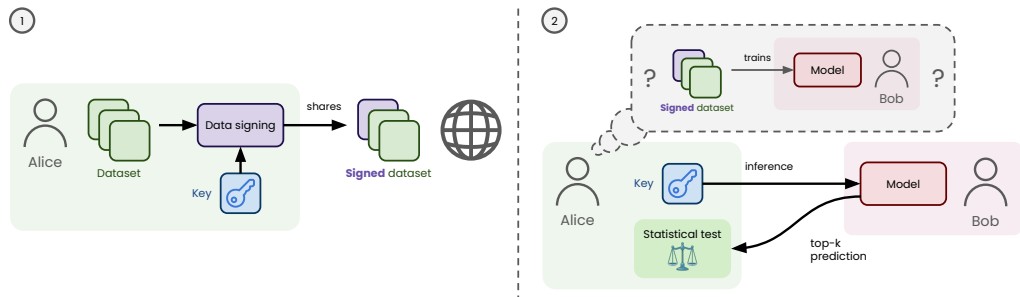

Figure 1: Application scenario of data taggants. ① Signing: Alice signs her dataset (adds the taggants corresponding to the keys) before publishing it. ② Detection: Alice determines if Bob used her dataset by running a statistical test based on Bob's model's predictions on the keys.

## 2 RELATED WORK

**Steganography and Watermarking.** Steganography is the practice of concealing a message within content, while watermarking is the process of marking data with a message related to that data (often as a proof of ownership) (Cox et al., 2007). They intersect in the contexts where watermarks strive to be imperceptible to avoid content degradation (Kahng et al., 1998). Recent works have shown that deep learning can improve watermarking technology (Vukotić et al., 2018; Zhu et al., 2018; Fernandez et al., 2022) and conversely, AI models can be watermarked (Adi et al., 2018). Watermarks are designed to be detected in data and their radioactivity on models (Sablayrolles et al., 2020; Sander et al., 2024) is only a byproduct. In contrast, data taggants are designed not to be detected, but to leave a mark on models that can later be detected with high confidence.

**Dataset ownership verification.** Our work, similarly to most works on dataset ownership verification (DOV), focuses on image classification datasets as their main use case. Therefore, we focus the discussion on this case. An early approach to DOV involved the modification of certain images in the dataset to align the last activation layer of a classifier with a random direction (Sablayrolles et al., 2020). This method demonstrated success in terms of stealthiness and provided robust theoretical guarantees in a white-box scenario, where access to the weights of the suspicious model is available. However, in the more realistic black-box scenario, the authors only proposed an indirect approach involving distillating the suspicious model, which requires a high number of queries [1]. More recently, attention has shifted towards backdoor watermarking for DOV (Li et al., 2020a; 2022; Wenger et al., 2022; Tang et al., 2023; Guo et al., 2023). These methods are closer to our black-box approach, although they require more information from the model and rely on confidence scores (Li et al., 2020a; 2022) rather than top-k predictions. The data is manipulated such that models trained on it alter their confidence scores when a trigger pattern is added to an input image. From a theoretical perspective, these approaches currently lack theoretical grounding. They all rely on the assumption that an honest model satisfies their null hypothesis – i.e. that a model not trained on the data should not change its confidence scores or predictions more than a predetermined threshold. The validity of the test is thus questionable, especially since they do not provide theoretical or empirical support for the choice of the threshold on confidence scores. In contrast, our approach precisely characterize the behavior of a benign model which enables the application of standard and sound statistical tests. Table 5 in Section A.1 in Appendix compares our work with prior approaches.

**Data poisoning.** Our research, akin to prior backdoor watermarking studies, repurposes data poisoning techniques for DOV. Data poisoning originally investigates how subtle changes to training data can compromise a model by malicious actors. Two main types of data poisoning attacks exist: backdoor attacks, which modify the model's behavior when a specific trigger is applied to a class of data (Li et al., 2019; Souri et al., 2021), and targeted attacks, which induce errors on a specific set of inputs (Shafahi et al., 2018; Geiping et al., 2020). These, in turn, may add incorrectly labeled data Li et al. (2020a; 2022) to the dataset, which impedes the stealthiness of the approach. Previous studies on backdoor watermarking were mostly based on backdoor attacks, attempting to predictably degrade performance when a trigger is introduced to the data. In contrast, we build on top of a *clean-label* targeted data poisoning approach (Geiping et al., 2020). We aim to design models that predict specific key labels in response to key inputs. The difference with a poisoning attack being that we alter the behavior on randomly generated patterns rather than legitimate data as to not induce malicious errors. We refer to that approach as *data signing*. Our algorithmic approach leverages gradient matching techniques developed in the clean-label data poisoning literature, for both targeted and backdoor attacks (Geiping et al., 2020; Souri et al., 2021). This technique allows to reproduce, via data poisoning, the effects of a particular gradient direction, which has been shown to be possible even for arbitrary gradient attacks Bouaziz et al. (2024).

**Membership inference attacks.** The goal of membership inference attacks is to reveal confidential information by inferring which data points were in the training set, usually recognized by low-loss inputs (Shokri et al., 2017; Watson et al., 2021). These methods do not offer any theoretical membership certificate, since a model might have low loss on a sample for different reasons than this sample simply being in the training set. MIAs are not applicable to DOV (Zhang et al., 2024).

---

[1] Black-box without distillation version of radioactive data amounts to a membership inference attack which hence lack the strong theoretical guarantees of the approach.

## 3 DATA TAGGANTS

In the application scenario we consider, Alice wants to publish online a dataset $\mathcal{D}_A$. Alice suspects Bob will try to train a model $\mathcal{M}^B$ on $\mathcal{D}_A$. Given **black-box** access (top-$k$ predictions) to Bob's model $\mathcal{M}^B$, Alice wants to mark her dataset in order to determine if $\mathcal{M}^B$ was trained on $\mathcal{D}_A$.

We propose a solution to Alice's problem called *data taggants*, which alters the dataset to mark models trained on it. Data taggants uses data poisoning to induce a certain behavior on Bob's model, and statistical tests to detect if Bob's model displays said behavior. To ensure **stealthiness**, we take inspiration from *clean-label* data poisoning, which leaves labels untouched (Geiping et al., 2020). Since the goal of our approach is to harmlessly influence the model, we designate our approach as *data signing* instead of data poisoning.

### 3.1 OVERVIEW AND TECHNICAL BACKGROUND

Let us denote Alice's original dataset by $\mathcal{D}_A = \{(x_i, y_i)_{i=1}^N\} \in (\mathcal{X} \times \mathcal{Y})^N$, where $N$ is the number of samples, $\mathcal{X}$ is the input space and $\mathcal{Y}$ is the set of possible labels. The process of adding data taggants is the following (Figure 1):

1. Alice generates a set of $K$ secret **keys**: $\mathcal{D}_{(key)} = \{(x_i^{(key)}, y_i^{(key)})_{i=1}^K\} \in (\mathcal{X} \times \mathcal{Y})^K$;

2. Alice **signs** (i.e. harmlessly poisons) her dataset by perturbating the images in a small subset $\mathcal{D}_S = \{(x_i^{(sign)}, y_i^{(sign)})_{i=1}^K\} \subseteq \mathcal{D}_A$ of size $S$ called the **signing set**. The perturbations $\Delta = \{\delta_i, i \in [S]\}$ added on top of images in $\mathcal{D}_S$ are called **signatures**, while **data taggants** refer to the modified signing set $x_i^{(taggant)} = x_i^{(sign)} + \delta_i, \forall i \in [S]$.

3. Alice shares $\tilde{\mathcal{D}}_A$, a modified version of $\mathcal{D}_A$ with the crafted data taggants replacing the elements in the signing set $\mathcal{D}_S$. The keys are kept **secret** and never shared with Bob.

The goal of the data taggants is to have models trained on $\tilde{\mathcal{D}}_A$ predict $y_i^{(key)}$ in response to $x_i^{(key)}$, while being **stealthy** so that Bob cannot easily remove it, and **robust** to different settings (model architecture, training algorithm) Bob could use. Let us denote by $\mathcal{L}_\theta^B$ the loss of Bob's model with parameter $\theta$. We use $t$ to denote a data augmentation sampled according to Bob's recipe and $\mathbb{E}_t[\cdot]$ the expectation over this random sampling. Alice defines a constrained set of image perturbations $\mathcal{C} = \{\delta \in \mathbb{R}^d / \|\delta\|_\infty \le \varepsilon\}$, where $\varepsilon > 0$ is kept small to ensure stealthiness. The space of possible signatures is then $\mathcal{C}_S = \{\Delta = (\delta_j)_{j=1}^N \in \mathcal{C}^N | \forall j \notin \mathcal{D}_S, \delta_j = 0\}$. Alice aims to find the signature $\Delta$ which minimizes the loss of Bob's model on the keys after training on $\tilde{\mathcal{D}}_A$. This corresponds to the following bilevel optimization problem:

$$\min_{\Delta \in \mathcal{C}_S} \sum_{i=1}^K \mathcal{L}_{\theta^*(\Delta)}^B(x_i^{(key)}, y_i^{(key)}) \quad \text{s.t.} \quad \theta^*(\Delta) \in \arg\min_\theta \frac{1}{N} \sum_{j=1}^N \mathbb{E}_t\left[\mathcal{L}_\theta^B(t(x_j + \delta_j), y_j)\right]. \quad (1)$$

Since solving the bilevel optimization problem above is intractable, we use a variant of gradient matching (Geiping et al., 2020), where the goal is to find $\Delta$ such that, when $\theta \approx \theta^*(\Delta)$,

$$\frac{1}{K} \sum_{i=1}^K \nabla_\theta \mathcal{L}_\theta^B(x_i^{(key)}, y_i^{(key)}) \approx \frac{1}{S} \sum_{j=1}^S \mathbb{E}_t\left[\nabla_\theta \mathcal{L}_\theta^B(t(x_j^{(sign)} + \delta_j), y_j^{(sign)})\right].$$

So that an optimization step on the data taggants also improves the model's loss on the keys. In reality, Alice does not know in advance Bob's training algorithm. There may also be multiple "Bobs" using different model architecture or training recipes. Alice thus cannot anticipate which architecture or training recipe to use to craft data taggants. Our approach involves a model and a set of data augmentation that Alice uses as a surrogate to Bob's training pipeline. We later study, in our experiments, the robustness of our approach when Alice and Bob use different model architectures or sets of data augmentations.

### 3.2 SIGNING THE DATA IN PRACTICE

The starting point of our approach is that Alice trains a model on her original dataset. We denote by $\theta^*$ the parameters of Alice's model, and $\mathcal{L}_{\theta^*}^A$ its loss function. We now give the details of the various steps to craft the taggants.

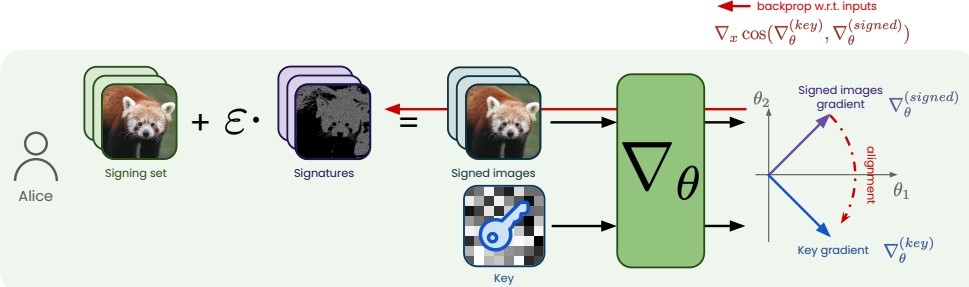

Figure 2: Illustration of our method: Alice optimizes image-wise signatures for images in the signing set by maximizing the alignment between the gradients of the signed images $\nabla_\theta^{(signed)}$, and the gradient of the key $\nabla_\theta^{(key)}$. The resulting images and their labels are the **data taggants**.

**Generating the secret keys.** To prevent from negatively impacting model performance, we choose key images $x_i^{(key)}$ to be **out-of-distribution** data points. More precisely, we generate key images by sampling each pixel value uniformly and sample key labels $y_i^{(key)}$ uniformly at random in $\mathcal{Y}$. Since no natural behavior is expected from the model on these keys, enforcing a specific behavior on them should not induce particular errors, as opposed to backdoor watermarking approaches.

**Parallelized and clean-label perturbations.** We evenly split the signing set into K parts $(\mathcal{D}_{S_i})_{i=1}^K$, where $\mathcal{D}_{S_i}$ is associated to key $i$. Following clean-label data poisoning (Geiping et al., 2020), each sample $(x_j, y_j)$ in $\mathcal{D}_{S_i}$ satisfies $y_j = y_i^{(key)}$. While Witches' Brew in the multi-targets case performs gradient matching to align the poisonous gradients with the *averaged targeted gradients*, we instead suggest to solve $K$ gradient matching problems between each part $\mathcal{D}_{S_i}$ and the associated key $(x_i^{(key)}, y_i^{(key)})$. The taggants optimization problem, described below, is thus decomposable into one problem for each key, which blue can be solved independently in parallel.

**Taggant objective function with differentiable data augmentations.** We use differentiable versions of data augmentations to optimize through them, similarly to Witches' Brew but increasing the set of considered data augmentations (see Table 13 in Appendix). At each gradient matching optimization step, we approximate $\mathbb{E}_t \left[ \nabla_\theta \mathcal{L}_{\theta^*}^A(t(x_j^{(sign)} + \delta_j), y_j^{(sign)}) \right]$ by resampling and applying $R$ randomly sampled data augmentations $t_r, r \in \{1, \dots, R\}$ (while $R = 1$ in Witches' Brew), and averaging the resulting gradients. The taggant objective function $\mathcal{T}(\Delta)$, illustrated in Figure 2, computes the alignment of perturbed images' gradients with the keys' gradients. More precisely, $\mathcal{T}(\Delta) = \sum_{i=1}^K \mathcal{T}_i / K$ with:

$$\mathcal{T}_i(\Delta) = -\cos \left( \underbrace{\nabla_\theta \mathcal{L}_\theta^A(x_i^{(key)}, y_i^{(key)})}_{\text{key gradient}}, \underbrace{\sum_{j \in \mathcal{D}_{S_i}} \frac{1}{R} \sum_{r=1}^R \nabla_\theta \mathcal{L}_\theta^A(t_r(x_j^{(sign)} + \delta_j), y_j^{(sign)})}_{\text{signed data gradients}} \right) \quad (2)$$

**Perceptual loss.** To improve the **stealthiness** of our approach, we *introduce* the use of a differentiable perceptual loss term $\mathcal{L}_{perc}$ to the taggant function, using the LPIPS metric (Zhang et al., 2018), which relies on the visual features extracted by a VGG model. Given a weight $\lambda$ for $\mathcal{L}_{perc}$, the final optimization problem for taggants is $\min_{\Delta \in \mathcal{C}_S} \frac{1}{K} \sum_{i=1}^K \mathcal{T}_i(\Delta) + \lambda \mathcal{L}_{perc}(\Delta)$.

**Random restarts.** Since the problem to solve is non-convex, the algorithm may be trapped in local minima. In practice, and similarly to Witches' Brew, we observed that using multiple random restarts of initial signatures for each key improved performance. We chose, for each key, the best crafted taggants among all restarts according to the taggant objective function.

## 3.3 DETECTION

The detection phase (②) in Figure 1) relies on checking a suspicious model's predictions on the secret keys. We consider the detection to be successful when a model displays a top-$k$ accuracy on the set of keys to be at least a threshold $0 \leq \tau \leq 1$. In order to control for false positives, we determine the top-$k$ accuracy of an honest model on the set of keys and set $\tau$ to be higher. The choice of $k$ and $\tau$ balances between the false positive rate (FPR, wrongfully detecting a benign model) and the true positive rate (TPR, correctly detecting a model trained on $\tilde{\mathcal{D}}_A$, i.e. detection rate). Given Bob's model's top-$k$ accuracy on the set of keys, Alice can derive a corresponding $p$-value for observing similar accuracy on a benign model. If that $p$-value is deemed low enough, it can be concluded that Bob's model was trained on Alice's dataset.

**Null hypothesis.** Our null hypothesis is $\mathcal{H}_0$: "model $h$ was not trained on the signed dataset".

**Proposition 1.** *Under $\mathcal{H}_0$, model $h$ has, in expectation, a top-k accuracy of $\frac{k}{|\mathcal{Y}|}$ on the set of keys $\mathcal{D}_K$, where $|\mathcal{Y}|$ is the number of possible labels.*

*Proof.* Under $\mathcal{H}_0$, for a given key, correctly predicting the random label $y^{(key)}$ based on a top-$k$ prediction given $x^{(key)}$ can be modeled by a random variable following a Bernoulli distribution of parameter $k/|\mathcal{Y}|$. Since all the labels are independently drawn, the number of correct predictions on the set of keys follows a binomial distribution with parameters $(K, k/|\mathcal{Y}|)$ and an expectation of $K \times k/|\mathcal{Y}|$. Hence the expected accuracy over $K$ keys is $k/|\mathcal{Y}|$. $\square$

**Statistical test.** Let us denote by top-$k(h, \mathcal{D}_K)$ the number of keys for which the label $y_i^{(key)}$ is in the top-$k$ predictions $h(x_i^{(key)})$ for $x_i^{(key)} \in \mathcal{D}_K$. Under the null hypothesis, Proposition 1 gives that top-$k(h, \mathcal{D}_K)$ follows a binomial distribution with parameters $(K, k/|\mathcal{Y}|)$ and can be subject to a binomial test. The $p$-value of the binomial test is then given by the tail of a random variable $Z_k$ following a binomial distribution $Bin(K, \frac{k}{|\mathcal{Y}|})$:

$$ p = \mathbb{P}(Z_k \geq \text{top-}k(h, \mathcal{D}_K)) = \sum_{z=\text{top-}k(h, \mathcal{D}_K)}^{K} \binom{K}{z} \left(\frac{k}{|\mathcal{Y}|}\right)^z \left(\frac{|\mathcal{Y}| - k}{|\mathcal{Y}|}\right)^{K-z} $$

This test only requires top-$k$ predictions, which are often available in black-box scenarios (e.g., restricted API access to the model).

Previous works on DOV have proposed to use a pair-wise t-test (Maini, 2021; Li et al., 2022; Wenger et al., 2022; Li et al., 2023; Tang et al., 2023; Guo et al., 2023) or a Wilcoxon signed-rank test (Li et al., 2023; Tang et al., 2023) on the model's predictions. However, the hypotheses these works test for rely (without any theoretical grounding) on the assumption that a benign model must display, on average, similar predictions on clean images and verification images, may they be watermarked images (Li et al., 2022; Wenger et al., 2022; Li et al., 2023; Tang et al., 2023) or particular private images (Maini, 2021; Guo et al., 2023). Our detection scheme, on the other hand, do not need any assumption as it allows us to exactly characterize the behavior of a benign model on the *keys*.

## 4 EXPERIMENTS

In this section, we empirically evaluate the effectiveness, stealthiness, and robustness of data taggants. Experiments are run on a realistic setting to assess the practicality of our approach.

**Experimental setup.** We use ImageNet1k (Deng et al., 2009) with Vision Transformers (DeIT) and Residual Networks (ResNet) models with different sizes and state-of-the-art training recipes from Wightman et al. (2021) and Touvron et al. (2022) (see Appendix A.3 for details). When generating taggants, the signing sets' sizes use a budget $B$ of the total dataset size, $S = B \times N$, with $B = 10^{-3}$ unless stated otherwise. We generate 20 secret keys, using 8 random restart per key and keep the $K = 10$ keys reaching the best taggant objective value. The weight of the perceptual loss is fixed to $\lambda = 0.01$. These parameters were chosen to produce visually imperceptible data taggants. $\varepsilon$ in the constraint set $\mathcal{C}$ is fixed to $16/255$, a common value in the data poisoning literature.

Table 1: Comparison of data taggants with DOV baselines when Alice and Bob train a DeIT-small on ImageNet1k with the Three Augment recipe with a budget of $0.1\%$. Aggregated over 4 runs, bold numbers indicate validation accuracy on par with clean training or better, and effective detection.

| Method | Validation accuracy | TPR | FPR | $\log_{10} p$ |
|---|---|---|---|---|
| Clean training | $\mathbf{64.2 \pm 0.4}$ | - | - | - |
| BW – Sleeper Agent | $\mathbf{64.4 \pm 0.3}$ | $0.0 \pm 0.0$ | $\mathbf{0.0 \pm 0.0}$ | $(0.0)$ |
| BW – BadNets | $\mathbf{63.7 \pm 0.5}$ | $0.0 \pm 0.0$ | $\mathbf{0.0 \pm 0.0}$ | $(0.0)$ |
| Data isotopes | $63.0 \pm 0.8$ | $0.53 \pm 0.09$ | $0.20 \pm 0.08$ | - |
| Data taggants (Our method) | $\mathbf{64.2 \pm 0.6}$ | $\mathbf{1.0 \pm 0.0}$ | $\mathbf{0.0 \pm 0.0}$ | $\mathbf{(-59.6)}$ |

Table 2: Validation accuracy and detection performance when both Alice and Bob use DeIT-small models with the three-augment data augmentation with a budget of $0.1\%$.

| Method | Validation accuracy | Top-$k$ keys accuracy | | | | PSNR |
|---|---|---|---|---|---|---|
| | | $k=1$ | $\log_{10} p$ | $k=10$ | $\log_{10} p$ | |
| Naive Canary | $63.8 \pm 1.1$ | $85.0 \pm 19.1$ | $(-91.7)$ | $\mathbf{100.0 \pm 0.0}$ | $\mathbf{(-74.0)}$ | - |
| Transparency | $63.6 \pm 0.6$ | $10.0 \pm 0.0$ | $(-4.9)$ | $\mathbf{55.0 \pm 5.8}$ | $\mathbf{(-29.7)}$ | $20.0$ |
| Data taggants (Our method) | $\mathbf{64.2 \pm 0.6}$ | $10.0 \pm 0.0$ | $(-4.9)$ | $\mathbf{87.5 \pm 5.0}$ | $\mathbf{(-59.6)}$ | $\mathbf{27.9}$ |

In each experiment, we train one model for Alice, craft data taggants, and train Bob's model on the now protected dataset. We run the detection test with $k = 10$ and $\tau = 0.1$ (for an associated theoretical FPR of $0.4\%$) and repeat each experiment 4 times with random initializations to compute standard deviations. Similarly to Sablayrolles et al. (2020), we combine the $p$-values of the 4 tests with Fisher's method (Fisher, 1970). The result is denoted $\log_{10} p$ and is commensurable to the base-10 logarithm of a $p$-value. Given the large number of experiments, we train all models for 100 epochs only, while a complete DeIT training is performed on 800 epochs. As sanity check, we confirm with a run of 800 epochs in Table 11 in Appendix A.2 the effectiveness of our method.

## 4.1 EFFECTIVENESS

We first compare the effectiveness of data taggants to three baselines (Section A.1 in Appendix explain this choice) for dataset ownership verification. First, **Backdoor watermarking** (BW) (Li et al., 2023), rely on a backdoor attack to embed a backdoor behavior in the model. To perform the backdoor attack, we leverage Sleeper Agent (Souri et al., 2021), an effective clean-label backdoor attack, and BadNets Gu et al. (2019), which applies a trigger on images from a source class and flips their labels to a target class. Finally, **Data isotopes** (Wenger et al., 2022), unlike BW, do not try to induce a backdoor behavior $f(x) = y \neq f(x + x^{(trigger)})$, but only to induce a slight change in the predicted logits. In this comparison, we use DeIT-small models with the Three Augment (3A) data augmentation and associated training recipe (Touvron et al., 2022) for Alice and Bob. We report the validation accuracies, the measured TPR and FPR ($\in [0, 1]$), and the $\log_{10} p$-value in Table 1.

Our experiments reveal that in a practical setting, BW is ineffective and yields a $0.0$ detection rate, and data isotopes offer a low detection rate of $0.53$ with a prohibitively high FPR of $0.20$ with a high toll on the validation accuracy. In contrast, data taggants achieve a *perfect* detection with a $1.0$ TPR and $0.0$ FPR, and a *very high confidence* of $p < 10^{-59}$, *without loss of performance*.

To measure the effectiveness of gradient matching to craft data taggants and force Bob's model to learn the keys, we compare it with two baselines to achieve the same goal. First, **"naive canary"**, where copies of the private keys are added into the training set. This serves as a topline in terms of detection performance, but is not viable as DOV mechanism due to its lack of stealthiness. Second, **"transparency"**, where we linearly interpolate between the keys and images of the signing set with a weight $\gamma = 0.2$, $x' = \gamma x^{(key)} + (1 - \gamma)x$. This value for $\gamma$ was chosen by visual inspection, as a

Table 3: Comparison of keys sources (test data *vs* random) when both Alice and Bob use DeIT-small models with the three-augment data augmentation and various budgets.

| Key source | Budget $B$ | Validation accuracy | Top-$k$ keys accuracy | | | | PSNR |
|---|---|---|---|---|---|---|---|
| | | | $k = 1$ | $\log_{10} p$ | $k = 10$ | $\log_{10} p$ | |
| None | 0.0% | $64.2 \pm 0.4$ | - | - | - | - | - |
| Test images | 0.001% | $63.7 \pm 0.4$ | $0.0 \pm 0.0$ | (0.0) | $7.5 \pm 5.0$ | (-1.1) | 27.5 |
| | 0.01% | $63.4 \pm 1.0$ | $0.0 \pm 0.0$ | (0.0) | $0.0 \pm 0.0$ | ( 0.0) | 27.9 |
| | 0.1% | $63.9 \pm 0.6$ | $0.0 \pm 0.0$ | (0.0) | $27.5 \pm 5.0$ | (-10.4) | 28.4 |
| Random (Our method) | 0.001% | $63.7 \pm 0.9$ | $0.0 \pm 0.0$ | (0.0) | $0.0 \pm 0.0$ | ( 0.0) | 26.6 |
| | 0.01% | $63.6 \pm 0.9$ | $0.0 \pm 0.0$ | (0.0) | $5.0 \pm 5.8$ | (-0.5) | 27.3 |
| | 0.1% | $64.2 \pm 0.6$ | $10.0 \pm 0.0$ | (-4.9) | $87.5 \pm 5.0$ | (-59.6) | 27.9 |

small value for which the key is still visible. In this comparison, we use DeIT-small models with the three-augment (3A) data augmentation (Touvron et al., 2022) for Alice and Bob, with three different budgets $B$, 0.1%, 0.01% and 0.001%.

The results are given in Table 2 (more results in Table 7 in Appendix). First, we observe that all methods have roughly the same validation accuracy as a model trained on clean data (differences within error). In particular, this suggests that data taggants do not negatively impact model performances at these budget levels. In terms of detection, Naive canary works best as expected, with perfect key top-10 accuracy for $B \geq 0.01\%$. While Transparency works better than data taggants for smaller budgets, data taggants has much higher top-10 accuracy for $B = 0.1\%$ (87.5% *vs* 55.0%). In addition, the PSNR (signal to noise ratio) is lower for transparency, which suggests that with a budget as small as 0.1%, data taggants already dominate the transparency baseline in this experimental setting, and can detect dishonest models with very high confidence ($\log_{10} p = -59.6$).

We perform an additional comparison with another baseline where test data points are used as keys rather than random patterns, which is similar to repurposing a data poisoning attack for DOV. Table 3 shows the results. With comparable PSNR, using random keys leads to much better detection accuracy and confidence for $B = 0.1\%$ than using test data, justifying our design choice.

## 4.2 STEALTHINESS

The stealthiness of data taggants is essential but difficult to measure beyond PSNR. As a best effort, we address the scenario where Bob tries to locate the signed data using either visual inspection, defense mechanisms against data poisoning, or, following (Tang et al., 2023), using anomaly detection algorithms. Similarly to the previous section, we use taggants crafted by DeIT-small using 3A augmentation, with a budget $B = 0.1\%$.

**Visual inspection.**  We provide examples of taggants crafted with and without perceptual loss in Figure 3 (more examples are given in Figure 6, 7 and 9 in Appendix A.2). We observe that while most of them appear unaltered, some data taggants display visible patterns of weak intensity that could as well come from natural film grain or compression artifacts. Although with PSNR < 30 dB (see Table 2) which is considered low according to image processing standards (lower than e.g. Sablayrolles et al. (2020)), we believe that the data taggants are hardly detectable via visual inspection, particularly when Bob has to find them among a whole dataset.

**Defense against data poisoning.**  Since our method relies on a data poisoning mechanism, we suggest to use data poisoning detection methods to detect data taggants. We consider three data poisoning detection methods relying on filtering samples: Deep k-NN (Peri et al., 2020), Activation Clustering (Chen et al., 2018) and Spectral Clustering (Tran et al., 2018). Over a wide range of parameters, the Receiver Operating Characteristic (ROC) curves in Figure 4 shows that data taggants cannot be detected by these methods with much better performances than random guess, suggesting strong stealthiness of data taggants.

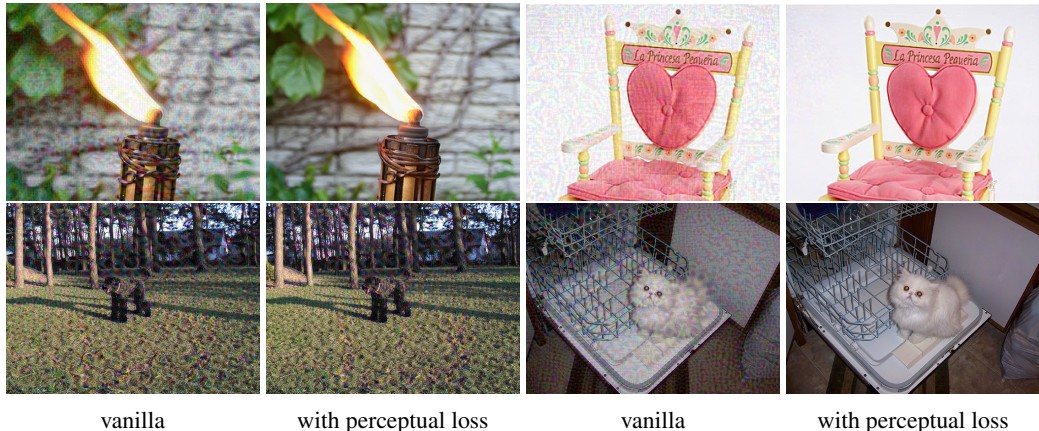

vanilla      with perceptual loss      vanilla      with perceptual loss

Figure 3: Pairs of data taggants crafted without perceptual loss (**left**) *vs* with perceptual loss (**right**).

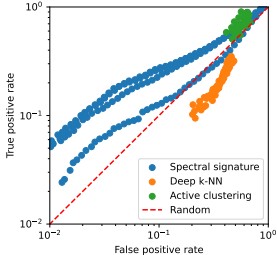

Figure 4: ROC curves for defense against data poisoning methods in the detection of data taggants.

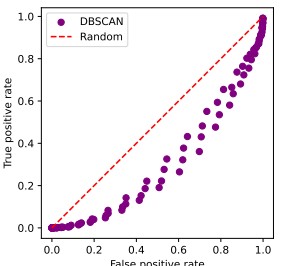

Figure 5: ROC curve for DBSCAN anomaly detection method.

**Anomaly detection.** Most methods for out-of-distribution (OOD) detection (Chen et al., 2020) rely on training a model on a set of in-lier data to then be able to detect outliers. In reality Bob does not have access to the original clean data nor to a benign model, hence cannot tell the outliers apart.

We consider the DBSCAN clustering method (Schubert et al., 2017) from the scikit-learn project[2] and run it with a wide range of thresholds ($\in [10, 35]$) to cluster the features of a model $h$ trained on $\tilde{\mathcal{D}}_A$. The resulting ROC curves are given in Figure 5 as a scatter plot after the experiment 4 times. Interestingly, when computing DBSCAN for different detection thresholds, we observe that it exhibit performances that are significantly worse than random. Our explanation is that DBSCAN computes clusters and select outliers as isolated points or clusters. By manually analyzing the clusters, we observe that some of them contain a lot of data taggants, most likely because their embeddings are lumped together and do not appear as anomalies.

On the one hand, these results suggest that OOD detection based on clustering is not a proper approach to detect the taggants. It suggests however that if Bob has sufficient resources, he may try to manually locate clusters of signed images by visual inspection, which is less costly than inspecting individual images. We leave this direction as an open question for future work.

### 4.3 ROBUSTNESS

For a fixed model and training algorithm, Table 2 and 3 already show the robustness of our method to a complete model retraining. But for our method to be practical, we need to ensure it will be robust when Bob's model or training algorithm is different from Alice's.

---

[2]https://scikit-learn.org/stable/index.html

Table 4: Robustness to data augmentation change. Alice does not know Bob's data augmentations and use either the Simple Augment or the Three Augment recipe.

| Bob's data aug. | Alice's data aug. | Validation accuracy | Top-$k$ keys accuracy | | | |
|---|---|---|---|---|---|---|
| | | | $k=1$ | $\log_{10} p$ | $k=10$ | $\log_{10} p$ |
| SA | SA | $58.1 \pm 0.3$ | $57.5 \pm 9.6$ | (-54.2) | $100.0 \pm 0.0$ | (-74.0) |
| | 3A | $56.1 \pm 0.3$ | $1.7 \pm 4.1$ | ( 0.0) | $15.0 \pm 8.4$ | ( -1.8) |
| 3A | SA | $64.1 \pm 0.6$ | $2.5 \pm 5.0$ | (-0.5) | $32.5 \pm 12.6$ | (-13.8) |
| | 3A | $64.0 \pm 0.5$ | $10.0 \pm 0.0$ | (-4.9) | $87.5 \pm 5.0$ | (-59.6) |

**Different data augmentations.** Table 4 (and Table 8 in Appendix A.2) shows how even when Alice does not know Bob's data augmentations, she can still successfully detect Bob's model, even though we gain much higher confidence when the same data augmentation is used by both Alice and Bob. Also, since the 3A data augmentation uses mixup (Zhang, 2017) and cutmix (Yun et al., 2019), these results also demonstrate the robustness of our approach to Borgnia et al. (2021) which showed that mixup and cutmix can be used to defend against data poisoning attacks.

**"Stress test": different model architectures and data augmentation.** We finally explore the robustness of data taggants in the most difficult setting, where Alice and Bob use different data augmentations as well as model architecture or sizes (see Table 12 in Appendix). We consider two families of models (DeIT and ResNet) of various sizes. We measure intra-family transferability by validating the protected dataset generated from each model onto each other model of the same family (DeIT and ResNet). We also measure inter-family transferability by crafting a protected dataset with ResNet-18 and DeIT-tiny models and validating it on all the models of the other family (resp. DeIT and ResNet). The results are shown in Table 9 in Appendix. Overall, we see good intra-family transferability, with larger DeIT models being more sensitive to taggants than smaller DeIT models. Interestingly, the trend is reversed for ResNets, with smaller ResNets being more sensitive. Across architectures, DeIT-tiny to ResNets or ResNets to DeIT-tiny, the results are less conclusive even though the top-10 accuracy is still $> 0$. All in all, these results suggest that the taggants are robust even in this worst-case scenario.

**Dataset change.** We explore the case where Bob trains his model on a modified version of Alice's dataset. We consider two cases: (1) Bob trains on multiple datasets (a superset of Alice's dataset), and (2) Bob trains on a subset of Alice's dataset. We present the results in Table 10 in Appendix and show data taggants are still effective in these cases.

## 5 LIMITATIONS AND FUTURE WORK

While our non-backdoor dataset ownership verification approach shows good properties in terms of *effectiveness*, *harmlessness*, *stealthiness*, and *robustness*, limitations are to be observed. Current results show a negligible degradation in validation accuracy compared to training on the clean dataset, which future works should try to further reduce. While we show the robustness of our method through different transfer experiments against different model architectures and training recipe, by modifying by modifying them, Bob can still hurt the detection performances. Obtaining higher confidence when Alice and Bob use different architectures and data augmentations is an interesting avenue for future work.

## 6 CONCLUSION

We introduced *data taggants*, a new approach for dataset ownership verification and designed mostly for image classification datasets. Data taggants are hidden in a dataset through gradient matching, in order to mark models trained on them. Our approach shows promising results, with very high detection rate and confidence, and low false positive rate, without affecting models' performance.

## 7 ETHICS STATEMENT

Our work is motivated by the need to ensure the integrity of machine learning models and the datasets they are trained on. While our approach to Dataset Ownership Verification (DOV) displays strong results, it is important to consider that such method can also fail. False positives can lead to misconceptions, particularly in such contexts.

## 8 REPRODUCIBILITY STATEMENT

The code implementing our method should be released upon publication. We provide all the necessary details to reproduce our experiments in the Section 4 and in the Appendix A.3.

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

# A APPENDIX

## A.1 COMPARISON WITH PRIOR WORK

To visually clarify the position of our work w.r.t. prior work, we represent them in Table 5 to highlight the main dimensions of comparison: (i) what are the means of detection, (ii) whether the method relies on black-box access to the model, (iii) whether it is clean-label, (iv) imperceptibility, and (v) what are the theoretical guarantees. A yellow cross ✗ indicates slight disagreement (e.g. for the 'black-box' axis of analysis, for method that use logits as mean of detection, since it represents significatively more information than just sharing the predictions). Please note that this table is coarse and represent the best case scenario for cases where a variety of methods can be used (e.g. Backdoor watermarking).

Among the prior work, we chose to discard several of them from our pool of baselines. Domain watermark (Guo et al., 2023) has yet to share the implementation and complete details for generating their "hardly-generalized domains" which limits us in reproducing their method. It is comparable to Data taggants but targets "hard" samples from their hardly-generalized domains that are deemed unlikely to be well classified if not trained on the protected data. On the other hand, we provide theoretical guarantees that the keys are indeed unlikely to be well classified by a benign model. Dataset inference (Maini, 2021) in black-box requires a very large amount of queries (a few hundred per data point to verify), which makes it impractical. Radioactive data (Sablayrolles et al., 2020) in black-box can only be done via distillation of the model to evaluate, which also require a large number of queries to the model. Data taggants, on the other hand, only requires $K$ queries ($K = 10$ in our experiments). We thus kept as baselines and compared against Backdoor watermarking and Data isotopes (Wenger et al., 2022). For Backdoor watermarking, we decided to leverage two approaches: a poisoned-label approach, BadNets (Gu et al., 2019), for its simplicity, and a clean-label approach, Sleeper Agent (Souri et al., 2021). The choice for this methods comes from the Domain Watermark (Guo et al., 2023) paper showing in Table 1 that Sleeper Agent performs better than the other tested clean-label backdoor watermarking approaches on Tiny-ImageNet.

| | mean of detection | black-box | clean-label | imperceptible | theoretical guarantees |
|---|---|---|---|---|---|
| Radioactive data (Sablayrolles et al., 2020) | weights/ logits | ✗ | ✓ | ✓ | on FPR |
| Dataset inference (Maini, 2021) | logits/ pred. | ✓ | ✓ | N/A | on linear models |
| Backdoor watermarking (Li et al., 2022; 2023) (Tang et al., 2023) | logits/ pred. | ✓ | not always | ✗ | ✗ |
| Data Isotopes (Wenger et al., 2022) | logits/ top-$k$ pred. | ✓ | ✓ | ✗ | ✗ |
| Domain Watermark Guo et al. (2023) | logits | ✗ | ✓ | ✗ | on risk |
| Data taggants (Ours) | top-$k$ pred. | ✓ | ✓ | ✓ | on FPR |

Table 5: Comparison of DOV methods along dimensions representing different desirable properties.

## A.2 ABLATIONS

**Visual inspections.** We show, in Figure 6, randomly chosen samples of data taggants generated with and without perceptual loss. The perceptual loss improves the stealthiness of the data taggants by making the perturbations less noticeable to the human eye. Some artifacts are still visible but could easily be overlooked by a human observer or misjudged as compression artifact or image grain.

**Comparison of the perceptual loss with weight decay.** We ensure that the gain in PSNR offered by the perceptual loss is not simply due to it reducing the perturbation amplitudes. Figure 7 compare data taggants generated with perceptual loss, with weight decay on the perturbation (replacing the perceptual loss term by the norm 2 of the perturbation $\|\delta_i\|_2^2$), and with their vanilla counterpart. It shows that for a similar PSNR, the weight decay version is much more noticeable than its perceptual loss counterpart. The perceptual loss hence plays an important role in hiding the signature.

**Clean performances.** We report clean performances in Table 6. This shows that the loss of accuracy is minimal when we train the model on data taggants.

Table 6: Validation accuracies for clean training of the different models and data augmentations used.

| Data aug. | Model | Validation accuracy |
|---|---|---|
| SA | DeIT-small | $56.1 \pm 0.5$ |
| 3A | DeIT-medium | $67.3 \pm 1.1$ |
| | DeIT-small | $64.2 \pm 0.4$ |
| | DeIT-tiny | $53.6 \pm 0.4$ |
| | ResNet-50 | $77.9 \pm 0.0$ |
| | ResNet-34 | $74.1 \pm 0.0$ |
| | ResNet-18 | $69.6 \pm 0.0$ |

**Baselines.** Similarly to our experiments, we report in Table 7, the validation accuracies and keys accuracies for our method and different baselines for keys uniformly sampled in the pixel space or from the test set. Figure 8 shows how Backdoor Watermarking-based detection (Li et al., 2023) performs with two backdoor attacks: Sleeper Agent (Souri et al., 2021) and BadNets (Gu et al., 2019). As Sleeper Agent displays a high $p$-value for both the watermarked model and the benign

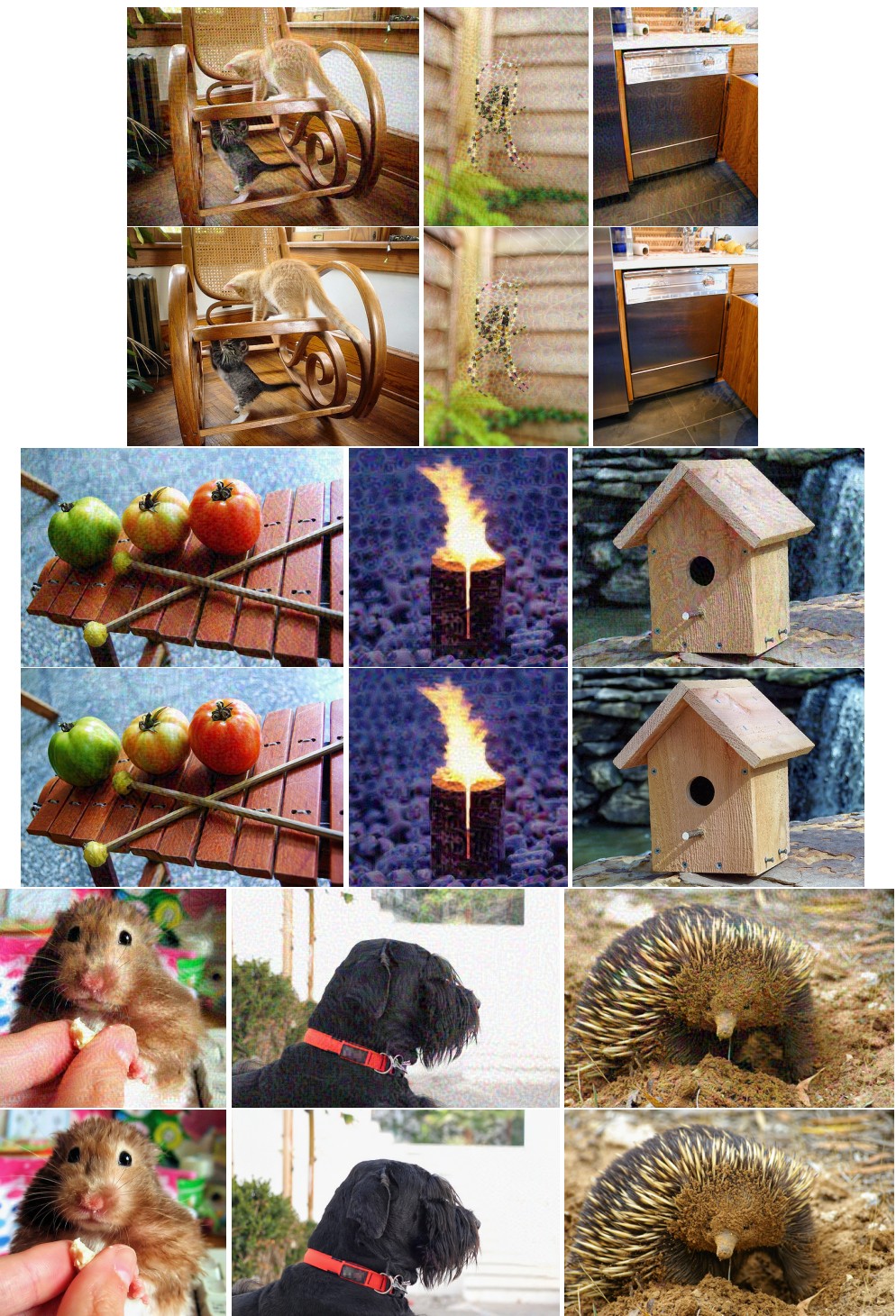

Figure 6: Comparison of data taggants generated without (**top**) and with perceptual loss (**bottom**). The images were sampled randomly.

model, any relevant level of significance (below $0.05$) yield low detection rate (True Positive Rate) and low false detection rate (False Positive Rate). On the other hand, as we lower the hyperparameter of the tested hypothesis, the $p$-value for associated with the benign model is lower than that of the watermarked model, which should lead to false detection rate higher than the detection rate. For

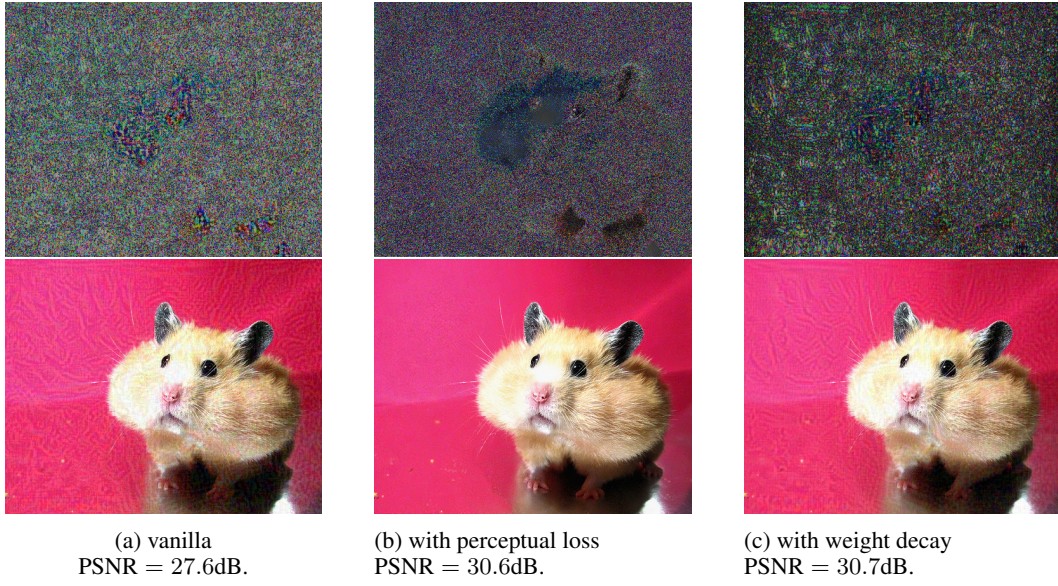

(a) vanilla
PSNR = 27.6dB.

(b) with perceptual loss
PSNR = 30.6dB.

(c) with weight decay
PSNR = 30.7dB.

Figure 7: Comparison of signatures (**top – amplified** $\times 10$) and data taggants (**bottom**) generated without discretion mechanism (**left**), with perceptual loss (**center**) and with weight decay (**right**).

BadNets, the $p$-value obtained from the t-test is lower for benign model than watermarked models as the threshold decreases, leading to a potentially high false detection rate.

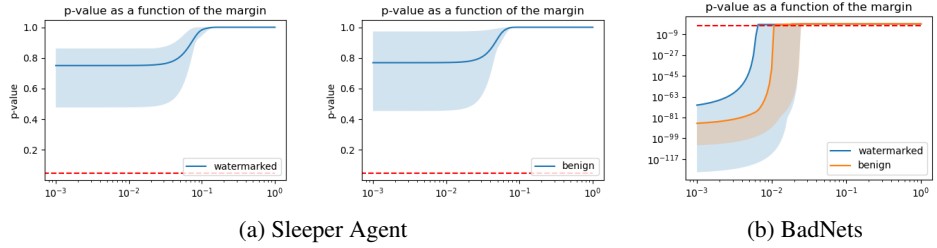

(a) Sleeper Agent

(b) BadNets

Figure 8: $p$-values computed for the Backdoor Watermarking-based detection (Li et al., 2023) as a function of the margin, the threshold of their null hypothesis.

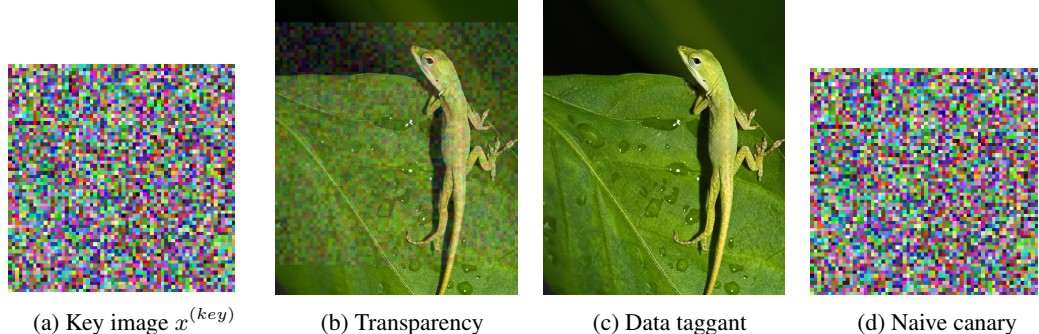

(a) Key image $x^{(key)}$     (b) Transparency     (c) Data taggant     (d) Naive canary

Figure 9: Comparison of data taggants with baselines "transparency" and "naive canaries" for a given key.

Table 7: Comparison of our data taggants for 10 keys against baselines for a ViT-small trained on ImageNet1k with the 3A recipe with various budgets. Averaged over 4 validation training runs each with different validation model's initialization. Errors represent the standard deviation.

| Key source | Method | Budget $B$ | Validation accuracy | Top-$k$ keys accuracy | | | | PSNR |
|---|---|---|---|---|---|---|---|---|
| | | | | $k=1$ | $\log_{10} p$ | $k=10$ | $\log_{10} p$ | |
| None | Clean training | 0.0% | $64.2 \pm 0.4$ | - | - | - | - | - |
| Test data | Naive Canary (Label Flipping) | 0.001% | $63.8 \pm 0.5$ | $0.0 \pm 0.0$ | (0.0) | $0.0 \pm 0.0$ | (0.0) | - |
| | | 0.01% | $64.2 \pm 0.7$ | $57.5 \pm 5.0$ | (-54.0) | $97.5 \pm 5.0$ | (-71.0) | - |
| | | 0.1% | $63.6 \pm 1.0$ | $100.0 \pm 0.0$ | (-113.4) | $100.0 \pm 0.0$ | (-74.0) | - |
| | Transparency | 0.001% | $64.2 \pm 1.1$ | $0.0 \pm 0.0$ | (0.0) | $0.0 \pm 0.0$ | (0.0) | 20.0 |
| | | 0.01% | $63.2 \pm 0.5$ | $0.0 \pm 0.0$ | (0.0) | $0.0 \pm 0.0$ | (0.0) | 20.0 |
| | | 0.1% | $63.7 \pm 0.6$ | $0.0 \pm 0.0$ | (0.0) | $0.0 \pm 0.0$ | (0.0) | 20.0 |
| | Data taggants | 0.001% | $63.7 \pm 0.4$ | $0.0 \pm 0.0$ | (0.0) | $7.5 \pm 5.0$ | (-1.1) | 27.5 |
| | | 0.01% | $63.4 \pm 1.0$ | $0.0 \pm 0.0$ | (0.0) | $0.0 \pm 0.0$ | (0.0) | 27.9 |
| | | 0.1% | $63.9 \pm 0.6$ | $0.0 \pm 0.0$ | (0.0) | $27.5 \pm 5.0$ | (-10.4) | 28.4 |
| Random data | Naive Canary | 0.001% | $63.4 \pm 0.7$ | $7.5 \pm 5.0$ | (-3.3) | $10.0 \pm 0.0$ | ( -1.8) | - |
| | | 0.01% | $64.2 \pm 0.3$ | $15.0 \pm 10.0$ | (-9.2) | $100.0 \pm 0.0$ | (-74.0) | - |
| | | 0.1% | $63.8 \pm 1.1$ | $85.0 \pm 19.1$ | (-91.7) | $100.0 \pm 0.0$ | (-74.0) | - |
| | Transparency | 0.001% | $63.5 \pm 1.0$ | $7.5 \pm 5.0$ | (-3.3) | $10.0 \pm 0.0$ | ( -1.8) | 20.0 |
| | | 0.01% | $63.4 \pm 0.7$ | $7.5 \pm 5.0$ | (-3.3) | $10.0 \pm 0.0$ | ( -1.8) | 20.0 |
| | | 0.1% | $63.6 \pm 0.6$ | $10.0 \pm 0.0$ | (-4.9) | $55.0 \pm 5.8$ | (-29.7) | 20.0 |
| | Data taggants (Our method) | 0.001% | $63.7 \pm 0.9$ | $0.0 \pm 0.0$ | (0.0) | $0.0 \pm 0.0$ | (0.0) | 26.6 |
| | | 0.01% | $63.6 \pm 0.9$ | $0.0 \pm 0.0$ | (0.0) | $5.0 \pm 5.8$ | (-0.5) | 27.3 |
| | | 0.1% | $64.2 \pm 0.6$ | $10.0 \pm 0.0$ | (-4.9) | $87.5 \pm 5.0$ | (-59.6) | 27.9 |

**Robustness to data augmentation changes.** We show in Table 8 that our method is robust to data augmentation changes. We also confirm that uniformly sampling keys in the pixel space to be out-of-distribution is a better strategy than using test data as keys.

Table 8: Robustness to data augmentation change. Alice does not know Bob's data augmentations and use either the Simple Augment or the Three Augment recipe. We compare our method with keys chosen from test data. $B = 10^{-3}$.

| Alice's data aug. | Bob's data aug. | Key source | Validation accuracy | Top-$k$ keys accuracy | | | |
|---|---|---|---|---|---|---|---|
| | | | | $k=1$ | $\log_{10} p$ | $k=10$ | $\log_{10} p$ |
| SA | SA | Random | $\mathbf{58.1 \pm 0.3}$ | $\mathbf{57.5 \pm 9.6}$ | **(-54.2)** | $\mathbf{100.0 \pm 0.0}$ | **(-74.0)** |
| | | Test data | $56.2 \pm 0.6$ | $15.0 \pm 17.3$ | (-9.9) | $60.0 \pm 14.1$ | (-34.2) |
| SA | 3A | Random | $\mathbf{64.1 \pm 0.6}$ | $\mathbf{2.5 \pm 5.0}$ | **(-0.5)** | $\mathbf{32.5 \pm 12.6}$ | **(-13.8)** |
| | | Test data | $63.9 \pm 1.1$ | $0.0 \pm 0.0$ | ( 0.0) | $2.5 \pm 5.0$ | ( -0.1) |
| 3A | SA | Random | $56.1 \pm 0.3$ | $1.7 \pm 4.1$ | ( 0.0) | $15.0 \pm 8.4$ | ( -1.8) |
| | | Test data | $56.1 \pm 0.4$ | $0.0 \pm 0.0$ | ( 0.0) | $\mathbf{16.0 \pm 5.5}$ | **( -3.9)** |
| 3A | 3A | Random | $\mathbf{64.0 \pm 0.5}$ | $\mathbf{10.0 \pm 0.0}$ | **(-4.9)** | $\mathbf{87.5 \pm 5.0}$ | **(-59.6)** |
| | | Test data | $63.9 \pm 0.6$ | $0.0 \pm 0.0$ | ( 0.0) | $27.5 \pm 5.0$ | (-10.4) |

**Robustness to stress test.** On top of the data augmentation changes, we show in Table 9 that our method displays some robustness to changes in the model architecture too.

**Dataset change.** We consider two cases:
(1) Bob trains on a superset of Alice's dataset. In our experiments, we make Alice use only half the classes of ImageNet1k and Bob trains on the whole dataset.
(2) Bob subsamples Alice's dataset. We make Alice use only half the classes of ImageNet1k. Bob will then remove 20% of Alice's samples from each class and add in the classes that do not belong to Alice.

Table 9: Stress test: Results of our method when Alice and Bob train two different architectures and different data augmentations. Alice uses SA data augmentations and Bob uses 3A.

| Bob's model | Alice's model | Validation accuracy | Top-$k$ keys accuracy | | | |
|---|---|---|---|---|---|---|
| | | | $k = 1$ | $\log_{10} p$ | $k = 10$ | $\log_{10} p$ |
| DeIT-medium | DeIT-medium | $67.2 \pm 1.3$ | $2.5 \pm 5.0$ | (-0.5) | $47.5 \pm 9.6$ | (-24.0) |
| | DeIT-small | $67.7 \pm 1.0$ | $5.0 \pm 5.8$ | (-1.7) | $60.0 \pm 8.2$ | (-33.8) |
| | DeIT-tiny | $67.6 \pm 0.7$ | $10.0 \pm 0.0$ | (-4.9) | $52.5 \pm 12.6$ | (-28.0) |
| | ResNet-18 | $67.0 \pm 0.8$ | $0.0 \pm 0.0$ | ( 0.0) | $10.0 \pm 0.0$ | ( -1.8) |
| DeIT-small | DeIT-medium | $64.5 \pm 0.6$ | $0.0 \pm 0.0$ | ( 0.0) | $22.5 \pm 9.6$ | ( -7.8) |
| | DeIT-small | $64.1 \pm 0.6$ | $2.5 \pm 5.0$ | (-0.5) | $32.5 \pm 12.6$ | (-13.8) |
| | DeIT-tiny | $63.7 \pm 1.0$ | $7.5 \pm 5.0$ | (-3.3) | $37.5 \pm 9.6$ | (-16.9) |
| | ResNet-18 | $64.3 \pm 0.6$ | $0.0 \pm 0.0$ | ( 0.0) | $10.0 \pm 0.0$ | ( -1.8) |
| DeIT-tiny | DeIT-medium | $53.8 \pm 0.4$ | $2.5 \pm 5.0$ | (-0.5) | $10.0 \pm 0.0$ | ( -1.8) |
| | DeIT-small | $53.9 \pm 0.5$ | $0.0 \pm 0.0$ | ( 0.0) | $12.5 \pm 5.0$ | ( -2.8) |
| | DeIT-tiny | $54.3 \pm 0.6$ | $2.5 \pm 5.0$ | (-0.5) | $12.5 \pm 5.0$ | ( -2.8) |
| | ResNet-18 | $53.5 \pm 0.5$ | $0.0 \pm 0.0$ | ( 0.0) | $7.5 \pm 5.0$ | ( -1.1) |
| ResNet-50 | ResNet-18 | $78.0 \pm 0.1$ | $5.0 \pm 5.8$ | (-1.7) | $15.0 \pm 5.8$ | ( -3.9) |
| | ResNet-34 | $77.9 \pm 0.1$ | $7.5 \pm 5.0$ | (-3.3) | $12.5 \pm 5.0$ | ( -2.8) |
| | ResNet-50 | $77.9 \pm 0.1$ | $2.5 \pm 5.0$ | (-0.5) | $37.5 \pm 27.5$ | (-18.5) |
| | DeIT-tiny | $77.8 \pm 0.2$ | $2.5 \pm 5.0$ | (-0.5) | $12.5 \pm 5.0$ | ( -2.8) |
| ResNet-34 | ResNet-18 | $74.2 \pm 0.1$ | $7.5 \pm 5.0$ | (-3.3) | $52.5 \pm 20.6$ | (-28.6) |
| | ResNet-34 | $74.1 \pm 0.1$ | $10.0 \pm 0.0$ | (-4.9) | $30.0 \pm 8.2$ | (-12.0) |
| | ResNet-50 | $74.1 \pm 0.1$ | $7.5 \pm 9.6$ | (-3.5) | $62.5 \pm 31.0$ | (-38.2) |
| | DeIT-tiny | $74.2 \pm 0.1$ | $10.0 \pm 0.0$ | (-4.9) | $10.0 \pm 0.0$ | ( -1.8) |
| ResNet-18 | ResNet-18 | $69.8 \pm 0.1$ | $7.5 \pm 5.0$ | (-3.3) | $55.0 \pm 31.1$ | (-32.0) |
| | ResNet-34 | $69.9 \pm 0.2$ | $7.5 \pm 5.0$ | (-3.3) | $22.5 \pm 15.0$ | ( -8.1) |
| | ResNet-50 | $69.8 \pm 0.1$ | $7.5 \pm 5.0$ | (-3.3) | $55.0 \pm 23.8$ | (-30.9) |
| | DeIT-tiny | $69.8 \pm 0.2$ | $5.0 \pm 5.8$ | (-1.7) | $10.0 \pm 0.0$ | ( -1.8) |

Table 10: Results of our method when Bob trains his model on a modified version of Alice's dataset.

| Case | Validation accuracy | Top-10 key accuracy | $\log_{10} p$ |
|---|---|---|---|
| (1) | $64.5 \pm 0.6$ | $62.5 \pm 15.0$ | $(-36.3)$ |
| (2) | $58.6 \pm 0.7$ | $72.5 \pm 9.6$ | $(-44.9)$ |

Results are showed in Table 10 and show that in both cases, our method is still able to detect Bob's model strong reaction to the keys.

**Result for 800 epochs training.** To ensure that the effects of our data taggants are still observed even during a full training, Table 11 reports the top-$k$ keys accuracies and associated p-values for a complete training of 800 epochs for random keys and keys taken from the test data when Alice and Bob both use the 3A data augmentation. Our method displays better performances of detection for the same validation accuracy.

Table 11: Results for a complete training of a deit-small for 800 epochs and comparison with keys chosen from test data. $B = 10^{-3}$.

| Key source | Validation accuracy | Top-$k$ keys accuracy | | | |
|---|---|---|---|---|---|
| | | $k = 1$ | $\log_{10} p$ | $k = 10$ | $\log_{10} p$ |
| Random data | $79.4 \pm 0.2$ | $\mathbf{10.0 \pm 0.0}$ | **(-4.9)** | $\mathbf{85.0 \pm 10.0}$ | **(-57.2)** |
| Test data | $79.4 \pm 0.3$ | $0.0 \pm 0.0$ | ( 0.0) | $0.0 \pm 0.0$ | ( 0.0) |

## A.3 Experimental details

**Computational resources.** All our experiments ran on 16GB and 32GB V100 GPUs. The different steps took the following time:

- Initial training: 14h-16h
- Data taggants generations: 2h-8h
- Validation training: 14h-16h

**Models and training recipes.** We present in Table 12 the list of models used and their size and number of parameters. Table 13 details the data augmentations used in our experiments. Finally, Table 14 presents the training recipe used for our experiments.

Table 12: Models and sizes.

| Model | # params. |
|-------|-----------|
| ViT-tiny | 5.46 M |
| ViT-small | 21.04 M |
| ViT-medium | 37.05 M |
| ViT-base | 82.57 M |
| ResNet-18 | 11.15 M |
| ResNet-34 | 20.79 M |
| ResNet-50 | 24.37 M |
| ResNet-101 | 42.49 M |

Table 13: Data augmentations.

| Data aug. | SA | 3A |
|-----------|-----|-----|
| H. flip | ✓ | ✓ |
| Resize | ✓ | ✓ |
| Crop | ✓ | ✓ |
| Gray scale | ✗ | ✓ |
| Solarize | ✗ | ✓ |
| Gaussian blur | ✗ | ✓ |
| Mixup alpha | 0.0 | 0.8 |
| Cutmix alpha | 0.0 | 1.0 |
| ColorJitter | 0.0 | 0.3 |
| Test crop ratio | 1.0 | 1.0 |

Table 14: Training recipe

| Family | DeIT | ResNet |
|--------|------|--------|
| Reference | (Touvron et al., 2022) | (Wightman et al., 2021) |
| Batch size | 2048 | 2048 |
| Optimizer | LAMB | LAMB |
| LR | $3.10^{-3}$ | $8.10^{-3}$ |
| LR decay | cosine | cosine |
| Weight decay | 0.02 | 0.02 |
| Warmup epochs | 5 | 5 |
| Dropout | ✗ | ✗ |
| Stoch. Depth | ✓ | ✓ |
| Repeated Aug | ✓ | ✗ |
| Gradient Clip. | 1.0 | 1.0 |
| LayerScale | ✓ | ✗ |
| Loss | BCE | BCE |

