# OpenReview forum: "Data Taggants: Dataset Ownership Verification Via Harmless Targeted Data Poisoning"
_ICLR.cc/2025/Conference — ICLR 2025 Poster_

### Official Review · Reviewer_CRSk · 2024-10-28

**Soundness:** 3
**Presentation:** 3
**Contribution:** 2
**Rating:** 8
**Confidence:** 5

**Summary:**

In this paper, the authors propose Data Taggants, a dataset ownership method used to detect unauthorized data usage. Data Taggants relies on clean-label targeted data poisoning technique and requires only black-box access to the suspected model. Data Taggants generate secret keys, i.e., (input label) pairs, and signed input samples by maximizing the alignment between keys and signed samples, and induce a certain behavior only on the models trained by the modified version of the dataset including those signed images. The verification procedure of Data Taggants include statistical tests using suspected model's top-k predictions on the secret key.

I think there is novelty, particularly considering the application, but an incremental one as Data Taggants use ideas from gradient matching (Geiping et al. 2020).

**Strengths:**

1. Empirical results show that Data Taggants have zero false-positive rate and high true positive rate while maintaining the model performance.
2. The generation of secret keys is purely random and is not included in the modified dataset, which makes key recovery almost impossible and unique to the data owner.
3. The presentation is clear.

**Weaknesses:**

1. As far as I understand, the verification includes querying the suspected model with keys. As they are purely random and out-of-distribution, the adversary might evade the verification by trying to detect those specific inputs and altering the predictions.
2. The method might be prone to watermark collusion: the adversary can generate its own key set and data taggants by modifying the already signed dataset, and after that it can also claim that the accuser is the malicious one.
3. Data Taggants has limited effectiveness and robustness when k=1 in top-k predictions.

**Questions:**

1. The radioactive data (Sabrayrolles et al., 2020) method has the option of black-box verification. In black-box verification, the radioactive data method compares the difference in loss between clean and radioactive images, and it does not necessarily involve training a student model to replicate the suspected model, it just checks the difference between the loss. Thus, authors' claim in page 1, line 053 as well as in page 3 lines 127-129 are incorrect. I strongly recommend changing the explanation.
2. I do not understand why the authors think that the independence of observations assumption does not hold in statistical testing. The models' predictions are independent of each other in the inference phase.
3. The authors empirically show that the data taggants are visually imperceptible, as designed in the methodology. It can work quite nicely on images with a large input space, but my question is how imperceptible this noise will be in data with lower-dimensional input spaces, e.g., smaller images like CIFAR10, gray-scale images or on a different data type like tabular data or text?
4.  In Table 1, the authors show that the backdoor watermarking (Li et al., 2023) has zero TPR and zero FPR. How authors measured such drastic numbers when the reference reports much better numbers? Is it because backdoor watermarking uses the full probability set instead of top-k labels or due to the mechanism of Wilcoxon-test?
5. What happens if the adversary decides to use a subset of the dataset? It will negatively affect the verification as the ratio of signed images to the whole dataset might decrease. Another case is how the performance of Data Taggants is affected when the adversary combines different datasets to train its model? The budget B will decrease and smaller budget produce worse results according to Table 3.
6. Page 2, line 148: typo while giving the reference

**Details Of Ethics Concerns:**

N/A.

---

> ### Author Response · Authors · 2024-11-21
> **Response to reviewer CRSk (1/2)**
>
> We thank the reviewer for the thorough and attentive review of our work, we highly appreciate the effort that was put in your review.
> Allow us to address the above-mentioned weaknesses:
> 1. This is an interesting point.\
>   First, Table 3 shows that our approach still works when replacing the random keys with test images (accounting for an actual data poisoning), other forms of keys could be experimented with, if needed, to avoid detection.\
>   Second, what OOD detection would you have in mind? Most of them (if not all) rely on having a set of inliers and a set of outliers. If you do not know beforehand what the outliers (here the keys) will look like, it is unclear how well it would work.\
>   Finally, such a countermeasure to data taggants could have downfalls on the provided service. The model provider would likely reduce the utility of their model to any user sending images that would be detected as being OOD.
> 2. We thank the reviewer for the relevant remark. Watermark/poisoning collisions can indeed be observed in classical settings targeting actual data points. By “collision”, we mean that a data poison (or backdoor watermark) supersedes the initial attack or gets a higher priority when training a model which can disable it.\
>   This can be the reason why targeted data poisoning can display difficulty to scale in terms of number of targets (e.g. Table 9 in [1] showing their method’s failure when targeting several images). Backdoor attacks however prove to be effective even in the case of multiple attacks as shown in [2].\
>   Our very method shows that several independent attacks can coexist, since we generate the data taggants for a given key independently from the others. To disable data taggants, an adversary would need to modify at least part of them. Given they only amount to 0.1% of the whole dataset (and already require a non-negligible amount of compute to be crafted), it would be really difficult for an adversary to find them and craft another poisoning on top of them.
> 3. The lower the $k$, the lower the measured top-$k$ accuracy, and the less effective our approach is. Top-$k$ prediction is still far less information required compared to other approaches (e.g. radioactive data). Also, one could mandate a model provider to give access to the top-$k$ prediction to their model to run the verification, protecting the model from being disclosed.
>
> Regarding your questions:
> 1. Radioactive data in black-box setting without distillation amounts to a membership inference attack and has none of the theoretical guarantees radioactive data approach offers in white-box or black-box distillation settings. Also, in the black-box without distillation setting, they only claim that "We can see that the use of radioactive data can be detected when a fraction of q = 20% or more of the training set is radioactive". Overall, radioactive data in black-box setting without distillation is a different approach that can hardly be said to work. We thank you for your remark and will make sure to change the description to explicit this distinction.
> 2. Their use of a t-test depends on the dataset that is chosen to run the test on, which in turn, becomes a factor of confusion. We will make sure to change the manuscript to clarify our criticism of backdoor watermarking’s statistical testing.\
>   One simpler point we would like to make is that the hypothesis they test for (Proposition 1 in [3] - $H_{1}: P_{b} + \tau < P_{w}$) has no theoretical grounding. As we show below, a benign model can as well display the same behavior they measure as a watermarked model.
> 3. The imperceptibility is mainly a matter of tradeoff between the gradient-matching loss and the perceptual loss. The dimension of the input space has a role in the effectiveness of the method. We found for instance that the method overall is less effective on CIFAR-10 but we can keep the perturbation relatively imperceptible with the perceptual loss weight. Other experiments show that the approach can be made both effective and imperceptible on 1-second 16kHz audio samples. Future work will be coming on other modalities.

---

> > ### Author Response · Authors · 2024-11-21
> > **Response to reviewer CRSk (2/2)**
> >
> > 4. The values presented in Table 1 for method [3] were obtained following the same detection protocol as the one they describe in their Algorithm 1, with a margin $\tau = 0.2$. When varying the margin and running the detection on 4 models, we observe a decreasing p-value that plateaus at 0.8 for a ridiculously low margin. The dashed red line is the 0.05 threshold of significance. Error bars represent max/min values.\
> >   [Plot image: p-value for the detection of a model watermarked with Sleeper Agent](https://i.postimg.cc/25yF3xy4/pval-margin-sleep.png)\
> >   When similarly running the detection on benign models (hence checking for potential false detection), we obtain the following results:\
> >   [Plot image: p-value for the detection of a benign model](https://i.postimg.cc/JzP51Frr/pval-margin-sleep-fpr.png)\
> >   This means that Sleeper Agent fails altogether on our setting.\
> >   We explain the discrepancy with what the authors reported in [4] by the differences in settings. Most notably, we use a far more challenging training recipe (with much more aggressive data augmentations). On the other hand, as another reviewer suggested we use the BadNet, we ran the same study on BadNet in the same experimental setting as Table 1 and obtained the following results on watermarked and benign models:\
> >   [Plot image: p-value for the detection of a model watermarked with BadNet and a benign model](https://i.postimg.cc/BbS3j1qD/pval-margin-badnet.png)\
> >   The p-value for the detection of benign models being lower than that of watermarked models indicates that running [3] detection algorithm can lead to a higher number of false positives than true positives.
> > The above-mentioned experiments show that backdoor watermarking in our setting either leads to low TPR and low FPR or high TPR and high FPR.
> > 5. Table 9 shows the performance of our method if Bob combines different datasets or subsamples Alice’s dataset. Cutting out data taggants indeed degrades the detection performance. However, since we only need a few hundred data taggants for the method to be effective, an adversary would need to trim a large number of samples to make sure to significantly reduce the detection performance. In our experiments, the degradation of the underlying model’s performance is important.
> > Also, please note that while the budget is certainly a consideration, it is ultimately the number of samples that it represents that holds greater significance.
> > 6. The typo has been corrected, thank you.
> >
> > We hope to have addressed all your concerns. We remain at your disposal may you have any further questions or require additional information. We would be grateful if you could consider revising your score based on the answers we provided.
> >
> > [1] Geiping, J., Fowl, L., Huang, W. R., Czaja, W., Taylor, G., Moeller, M., & Goldstein, T. (2020). Witches' brew: Industrial scale data poisoning via gradient matching.\
> > [2] Alex, N., Siddiqui, S. A., Sanyal, A., & Krueger, D. (2024). Protecting against simultaneous data poisoning attacks.\
> > [3] Li, Y., Zhu, M., Yang, X., Jiang, Y., Wei, T., & Xia, S. T. (2023). Black-box dataset ownership verification via backdoor watermarking.\
> > [4] Souri, H., Fowl, L., Chellappa, R., Goldblum, M., & Goldstein, T. (2022). Sleeper agent: Scalable hidden trigger backdoors for neural networks trained from scratch.

---

> > > ### Comment · Reviewer_CRSk · 2024-11-25
> > > **Official Comment by Reviewer CRSk**
> > >
> > > Thank you for your detailed response and clarification of radioactive data.
> > >
> > > It is true that most state-of-the-art OOD detection methods rely on training the detection module using both in- and out-of-distribution samples. The adversary in this case can collect any OOD samples or generate random noise. Of course, this would affect the overall utility, but then also could serve as the simplest evasion technique against Data Taggants.
> > >
> > > I will keep my score as it is.

---

> ### Author Response · Authors · 2024-11-25
>
> Thank you for engaging in the discussion.
>
> It is not straightforward OOD detection can detect keys when Bob has no idea what the keys might even look like (i.e. if Bob has no idea what the distribution of the keys might be). While we experimented with OOD keys where pixels are uniformly sampled, Table 3 shows that the method *still work for in-domain data*. Follow-up work can consider many form of keys and explore which ones are the most effective and stealth to OOD detection.
>
> It appears to us that we addressed all your concerns. What could make you reconsider and increase your score?

---

> > ### Comment · Reviewer_CRSk · 2024-11-28
> > **Official Comment by Reviewer CRSk**
> >
> > Thanks authors for responses.
> >
> > I have checked the modified version of the submission, and throughly analyzed whether other reviewer's concerns were addressed in the revised version. I belive the revised version has a more solid discussion of prior work, as well as Witches' Brew, and with additonal experiments. That's why I am willing to increase the score.

---

> > > ### Author Response · Authors · 2024-11-28
> > > **Thank you for your thorough analysis**
> > >
> > > We thank you for engaging in the discussion. And we particularly thank you for taking the time to analyze our updated manuscript not only to make sure we addressed your concerns but also for the other reviewer's concerns and questions as well.
> > >
> > > We are glad to see you deemed our revisions convincing and updated your score accordingly.\
> > > We are also willing to discuss any other related point you would like for the remaining week of discussion.
> > >
> > > Best regards,\
> > > The authors

---

### Official Review · Reviewer_RFB3 · 2024-11-01

**Soundness:** 2
**Presentation:** 2
**Contribution:** 2
**Rating:** 5
**Confidence:** 4

**Summary:**

The paper proposed a dataset ownership verification method that can work in a black-box setting, where model weights and training details are not known in advance; Besides, the method is also stealthy compared to the backdoor-based method since it only requires limited perturbations to the dataset; Moreover, the method is also less harmful than the previous backdoor-based method.

**Strengths:**

- The proposed method focuses on harmlessness, stealthiness, and black-box, which are three important challenges in the ownership verification problem.
- The writing is easy to understand and follow.

**Weaknesses:**

- Novelty issues: Could you compare this paper with [1] in more detail, such as technical details, setting, and problem setup? Since in my understanding, [1] used a similar gradient-matching-based method to find some "hardly generalized domain", which is very similar to this method on a high level.
- Unclarified arguments: In Lines 59-60, the authors mentioned that 'but is also harmful to the model as it introduces errors [1]'. Could you further clarify what kind of errors the backdoor-based method introduces? In my personal understanding, the claim of "harmless" in [1] is mainly based on the fact that the backdoor-based method will leave exploits in the dataset, which will then further be maliciously used by the adversaries.
- Unclarified intuitions: The intuitions on why the "out-of-distribution" samples are used to construct key images are not further clarified.
- Experimental Details: Why do you choose SleeperAgent as the backdoor method for the baseline "Backdoor watermarking"? SleeperAgent is not the simplest way to inject backdoors and even requires an additional surrogate model to optimize perturbation $\delta$ to the original dataset. Therefore, could you (1) further clarify what is the necessity of choosing SleeperAgent, (2) provide more explanations on why the backdoor watermarking only achieves 0 TPR on your setting, and (3) provide additional experiments on the Backdoor watermarking with BadNet?

[1] Junfeng Guo, Yiming Li, Lixu Wang, Shu-Tao Xia, Heng Huang, Cong Liu, and Bo Li. Domain watermark: Effective and harmless dataset copyright protection is closed at hand.

**Questions:**

See the weakness part.

---

> ### Author Response · Authors · 2024-11-21
> **Response to reviewer RFB3 (1/2)**
>
> We thank the reviewer for their review of our work.
> To address the above-mentioned weaknesses:
> - **_Regarding the novelty:_**\
>   [1] relies on creating **samples that should be difficult to classify** when a model is trained on a benign dataset but easy to classify when trained on the protected dataset. Our approach, on the other hand, relies on randomly sampled out-of-distribution **samples that have no ground truth label**, and then force the model to make a decision on these samples when trained on our protected dataset.\
> Even though [1] also uses gradient matching to craft the perturbations, it is used after they trained a _domain adaptation model to generate their “domain watermarked samples”_, which is the main contribution of their work. That part (subsection 3.3 in [1]) is not straightforward and requires training a model to generate “new domain” samples.\
> The authors of [1] open sourced part of their code on Sept. 16th of this year, without the code to train a domain adaptation model, making their work extremely difficult to reproduce as such.
>
> - **_Regarding the harmfulness argument:_**\
>   Caption of Figure 1 in [1] states that:
> > "Existing backdoor based methods make the watermarked model (i.e., the backdoored DNN) misclassify ‘easy’ samples that can be correctly predicted by the benign model and therefore the verification is harmful"
>
>   Given a sample $x$ which is correctly classified as $y_{true}$, adding the trigger signal $t$ makes the watermarked sample $x+t$ which is supposed to be classified as $y_{w} \neq y_{true}$ by a watermarked model. Given that a trigger only makes sense if it does not alter too much of the data, a cleanly trained model should classify $x+t$ as $y_{true}$ (e.g. a $224 \times 224$ fish picture with a $16 \times 16$ patch in the corner is reasonably still a fish picture). Hence, the backdoor is here to introduce errors.\
>   This is exactly what [1]'s measures of harmfulness (Harmful $H$ and Relatively Harmful Degree $\hat{H}$ in Definition 1) measure and what their argument is based on.
>
> - **_Regarding the intuition behind the use of out-of-distribution samples as keys:_**\
>   As we explain on line 235:
>   > "Since no natural behavior is expected from the model on these keys, enforcing a specific behavior on them should not induce particular errors, as opposed to backdoor watermarking approaches."
>
>   Could you clarify what you would like us to add? Would you be ok with the following rephrasing:
>   > "Since no natural behavior is expected from the model on these keys, as they are made of random pixels, enforcing a specific behavior on them should not induce particular errors, as opposed to backdoor watermarking approaches, which relies on modifying the behavior of a model on actual images."
>
> - **_Regarding choosing Sleeper Agent:_**\
>   Even though Sleeper Agent is not the simplest backdoor attack, it is an effective approach. Please notice that Table 1 also includes “Data isotopes” [2] which uses a more traditional backdoor watermarking approach relying on blending a visible trigger into the image.\
>   (1) Because Sleeper Agent similarly leverages gradient-matching, it allows us to compare fairly the backdoor watermarking approach which uses triggers against data taggants which uses keys.\
>   (2) The values presented for method [3] were obtained in our setting and following the same detection protocol as the one described in Algorithm 1 of [3], with a margin $\tau = 0.2$. When varying the margin and running the detection on 4 models, we observe a decreasing p-value that plateaus at 0.8 for a ridiculously low margin. The dashed red line is the 0.05 threshold of significance. Error bars represent max/min values.\
>   [Plot image: p-value for the detection of a model watermarked with Sleeper Agent](https://i.postimg.cc/25yF3xy4/pval-margin-sleep.png)\
>   When similarly running the detection on benign models (hence checking for potential false detection), we obtain the following results:\
>   [Plot image: p-value for the detection of a benign model](https://i.postimg.cc/JzP51Frr/pval-margin-sleep-fpr.png)\
>   This means that Sleeper Agent fails altogether on our setting.\
>   We explain the discrepancy with what the authors reported in [4] by the differences in settings. Most notably, we use a far more challenging training recipe (with much more aggressive data augmentations).\
>   (3) As requested, we ran the same experiments (same model, dataset, poisoning budget) using the same detection method (Algorithm 1 in [1]) using the BadNet backdoor attack and obtained the following results on watermarked and benign models:\
>   [Plot image: p-value for the detection of a model watermarked with BadNet and a benign model](https://i.postimg.cc/BbS3j1qD/pval-margin-badnet.png)\
>   The p-value for the detection of benign models being lower than that of watermarked models indicates that running [3] detection algorithm can lead to a higher number of false positives than true positives.\

---

> > ### Author Response · Authors · 2024-11-21
> > **Response to reviewer RFB3 (2/2)**
> >
> > The above-mentioned experiments show that backdoor watermarking in our setting either leads to low TPR and low FPR (for Sleeper Agent) or high TPR and high FPR (BadNet).
> >
> > Also, a very important question arises on our side regarding your flag of our work for ethics review. This flag is an important assertion and we would very much appreciate if you **could provide arguments as for the reason for flagging our work for Ethics Review?**
> >
> > We hope to have addressed all your concerns and would be grateful if you could revise your score in return. We would be glad to address any further question otherwise.
> >
> > [1] Junfeng Guo, Yiming Li, Lixu Wang, Shu-Tao Xia, Heng Huang, Cong Liu, and Bo Li. (2024) Domain watermark: Effective and harmless dataset copyright protection is closed at hand.\
> > [2] Wenger, E., Li, X., Zhao, B. Y., & Shmatikov, V. (2022). Data isotopes for data provenance in DNNs.\
> > [3] Li, Y., Zhu, M., Yang, X., Jiang, Y., Wei, T., & Xia, S. T. (2023). Black-box dataset ownership verification via backdoor watermarking.\
> > [4] Souri, H., Fowl, L., Chellappa, R., Goldblum, M., & Goldstein, T. (2022). Sleeper agent: Scalable hidden trigger backdoors for neural networks trained from scratch.

---

> ### Author Response · Authors · 2024-11-25
> **Please consider engaging in the discussion**
>
> We thank the reviewer for their review and encourage you to engage in the discussion, replying to our rebuttal.
>
> We have carefully addressed the main concerns in detail and updated the manuscript accordingly.
> Is there any remaining concern before you can consider increasing your score? We would be glad to clarify any further concerns (if any)
>
> Best regards,\
> Authors

---

### Official Review · Reviewer_3ipm · 2024-11-03

**Soundness:** 2
**Presentation:** 3
**Contribution:** 3
**Rating:** 6
**Confidence:** 4

**Summary:**

The paper proposes an active dataset ownership verification (DOV) method, by adapting a technique for targeted data poisoning from prior work. The key advantages of the proposed approach are its applicability given only top-k black-box access, more principled/rigorous statistical certificates compared to prior work, stealthiness, and robustness to different setups as well as explicit defenses. All these properties are validated via thorough experimental evaluation.

**Strengths:**

- DOV is an important problem, and authors explicitly focus on realistic setups and often overlooked aspects such as the rigor of stated guarantees that accompany methods.
- The method is original within the space of DOV. The idea of repurposing witches brew and introducing random data sampling to strengthen the theoretical guarantees is very interesting and unexpected.
- Evaluation focuses on a large-scale practical setup and addresses many important points, evaluating stealthiness and robustness explicitly. I appreciate the inclusion of poisoning defenses and OOD detection.
- Setting the scope of evaluation aside (see below), the provided results seem quite strong.
- The paper is mostly well-written and easy to read, with some exceptions discussed below.

**Weaknesses:**

I can identify several important weaknesses of the work in its current state, and provide suggestions how these could be improved:
- **Incomplete evaluation/related work positioning**: While DOV is a crowded space and many baselines are cited in the paper, only two are run in the experimental part, without clear rationale, and the relationship to prior methods is in my view not clearly presented in the paper. For example, while the position of the paper seems to be "there may be DOV methods with strictly better TPR but they come with problems such as unrigorous guarantees or perceptible data changes", current Table 1 shows Taggants are the best even when only measuring TPR, which to me suggests that baselines are missing. The field is complex and there are many dimensions (active vs passive, blackbox top-k vs needs logits vs needs whitebox, different guarantee types, clean label vs perceptible, etc.). To give clarity, I believe the paper must (i) clearly outline all dimensions and place all prior baselines within them (ii) include any viable baseline (e.g., a perceptible method can be still run to demonstrate that even though it achieves high FPR, it fails a data poisoning defense) and clearly state why the others can not / should not be included. This would greatly improve the trust in the experimental results and make the case for Taggants.
- **Unclear claims of technical contribution**: The paper should clearly mark that many technical parts are directly lifted from Witches Brew (e.g., augmentations, restarts), while some other parts are introduced by this work (e.g., the use of random data, perceptual loss). The current writing can easily be interpreted as an overclaim, esp. by a reader not familiar with prior work. The actual contributions are quite interesting, and I do not think the lack of tech. contribution is a weakness of the paper in any case.
- **Unsubstantiated claims around guarantees**: One of the key claimed advantages of Taggants are rigorous guarantees not offered by prior work, as (i) random data samples are actually independent and (ii) under the null, the classifications of random data are actually uniform. While I tend to agree on an intuitive level, I believe (1) the reasons why prior work violates (i,ii) could be more clearly explained, e.g., ln301 simply states that "using model's predictions on ground truth class" violates the independence assumptions, but does not elaborate; (2) to show actual impact of this oversight of prior work, it should be empirically demonstrated that there is a mismatch between theoretical and empirical FPR (3) for taggants, there should be a corresponding matching FPR empirical validaiton, and a more detailed discussion around why taggants do not break the assumptions. Are model predictions on random [0,1]^d data really uniformly random? All these images are unusually high-variance compared to natural data; if we had a class such as "TV static" I can imagine they would all be classified as such? Do we need a different OOD distribution in this case, and how would we choose it? This needs more clarity as it is uite central to the paper.
- **[Minor] Key technical contribution undexplored**: If I understand correctly, the motivations given for how Keys are sampled are more rigorous guarantees as above, and lower likelihood to alter model utility, as data is OOD. However, Table 3 also shows forcing the model to predict a certain class is easier in this case than for in-distribution test images. If am not misinterpreting Table 3, it would be interesting to know why this is the case, and state it as the third reason for using such Key sampling to avoid confusion. Is it that gradient matching is here a better proxy for the true objective, or the objective is easier to optimize as we are far from the real data manifold? This seems underexplored but is a central idea of the paper.

Typos and points that do not affect my evaluation:
- ln151: dot missing, ln518: extra dot, ln188: extra "them". ln317: "In each experiment..." sentence seems wrong, not sure where.
- Related work says "[Data/model] watermarks are not designed to persist through processes that use the data", but I am not sure this is really the case, as these watermarks are generally designed with the goal of robustness. There are works that show (albeit on text) explicitly that such watermarks can persist through processes of finetuning and RAG (see Sander et al. "Watermarking Makes Language Models Radioactive" and Jovanovic et al. "Ward: Provable RAG Dataset Inference via LLM Watermarks")---this discussion could be included to give context. On a similar note, the data/model/backdoor watermarks distinction could be made clearer, e.g. by changing the first paragraph title in Sec. 2.

I am happy to hear from authors regarding these points and discuss them further.

=====

UPDATE: Score increased from 5 to 6 after rebuttal; see discussion thread below.

**Questions:**

- Optimization is done only w.r.t. fully trained model parameters. Yet, the goal of the gradient matching is to make training a model from scratch on Taggants equivalent to training it on Keys. Why are some randomly initialized models not included? Do you have insight why despite this, the surrogate objective seems to work?
- How should tau=0 on ln317 be interpreted? If I understand correctly, this means all models with non-zero accuracy on Keys are flagged?

---

> ### Author Response · Authors · 2024-11-20
> **Response to reviewer 3ipm (1/2)**
>
> We thank the reviewer for the thorough and attentive review of our work, we highly appreciate the effort that was put in your review.
> We would like to first address the above-mentioned weaknesses:
> - **_Regarding the evaluations and related works positioning:_**\
>   In your review, you say:
>   > the position of the paper seems to be "there may be DOV methods with strictly better TPR but they come with problems such as unrigorous guarantees or perceptible data changes"
>
>   Could you please elaborate on the elements that made you believe this was the position of our paper? Especially given that we already show in Table 1 that our method achieves better TPR than baselines.\
>   We added another baseline in the very same setting as Table 1 that we initially discarded: _Backdoor watermarking using the BadNet approach_ (a visible fixed trigger) and the detection method from [3] on 4 watermarked models (to compute the TPR) and 4 benign models (to compute the FPR). This approach relies on an additional hyper-parameter $\tau$ that controls the sensitivity of the test:\
>   [Plot image: p-value for the detection of a model watermarked with BadNet and a benign model](https://i.postimg.cc/BbS3j1qD/pval-margin-badnet.png)\
>   Given that the p-value of the benign models is lower than that of the watermarked models, for any threshold of significance, BadNet would lead to a higher FPR than TPR in this example, making the method _unreliable_.\
>   To improve the clarity of the comparison with previous work, we plan to _add a table to the paper to draw a comparison between our work and the related works across the relevant dimensions of comparisons_ and further explain which baselines we found to be relevant to compare against and which ones were discarded. We hope that would be sufficient to address your concern.
> - **_Regarding the claims of technical contribution:_**\
> We thank the reviewer for taking our contributions into consideration. We believe the current version of the paper already lists the contributions at the end of the introduction. We nonetheless would like to address your point and will update the manuscript to properly highlight our contributions in the rest of the paper.
> - **_Regarding our theoretical guarantees:_**\
> We need to clarify that **we do not make any assumption on the classifications/predictions of models on the keys** (randomly sampled OOD data points). Regardless of the model, if it is benign, it was not exposed to information about the keys (i.e. either the keys or the data taggants). Because the keys’ labels are random, *accuracy on random labels can only amount to chance level*. To detail what was shown in the proof of the Proposition 1:
>   - the *accuracy* of a benign model on *1 key* must follow a Bernoulli distribution with parameter $\frac{1}{|\mathcal{Y}|}$;
>   - hence the *top-$k$ accuracy* on *1 key* must follow a Bernoulli distribution with parameter $\frac{k}{|\mathcal{Y}|}$;
>   - since the labels of the $K$ keys are sampled independently, the *number of correct top-$k$ predictions* on the *$K$ keys* follows a binomial distribution with parameters ($K$, $\frac{k}{|\mathcal{Y}|}$).
>
>   This allows us to have a theoretical FPR for any observed performance displayed by a model. Given its level (as low as $10^{-60}$), we unfortunately cannot empirically validate it as it would require us running at least thousands of measures to expect one of them to be a false positive. Each of these measures requires training a model from scratch on ImageNet1k (each of them requiring roughly 200 GPU-hours). This would amount to an unreasonably large compute time, making empirical validation of the FPR infeasible. After running our detection procedure on a dozen models, we found a FPR of 0 as reported in Table 1. Backdoor watermarking, on the other hand, cannot provide any theoretical guarantee on the FPR because they cannot characterize the expected behavior of a benign model in their setting.
> Your remark on the choice of the OOD samples is also relevant. If there was a “TV static” class, then the keys we used in our experiments would hardly be OOD anymore and would amount to choosing the keys among the test images (from the “TV static” class) as shown in Table 3.
> - **_Regarding the exploration of the key technical contribution:_**\
> The exploration you mention would be interesting but falls in a much broader study about gradient matching which seems out of the scope of this paper. Future work on understanding training dynamics should definitely consider addressing this question.
>
> We thank you very much for noticing typos and we made sure to correct them in the manuscript right away.

---

> ### Author Response · Authors · 2024-11-20
> **Response to reviewer 3ipm (2/2)**
>
> Allow us to address your questions:
> 1. The reason why gradient matching works even when only crafting the gradients from a fully trained model is still not understood. [1] suggest to retrain the model during the poison crafting to avoid overfitting to a clean-trained model. This approach induces high training cost when dealing with large-scale datasets such as ImageNet1k.\
> The idea of introducing randomly initialized models when crafting poisons have not been explored to the best of our knowledge. Some experimental results we obtained when reproducing witches’ brew [2] experiments on CIFAR-10 showed that the more trained Alice’s model is, the better the poisoning works.\
> Our intuition is that neural networks could be using similar features, even at different initializations (and even architecture as per our stress-test experiments). As such, optimizing the data taggants on Alice’s trained surrogate model is enough to have features emerging in the data that can be learned as expected by a newly initialized model. Conversely, when crafting data taggants from a poorly trained model, because the feature extractor has yet to fully emerge, it fails to properly allow to derive relevant features that can be transferred to different models.
> 2. You are right. Here, in our experiments, we consider any non-zero accuracy on the keys to be suspicious, which leads to a 100% TPR and 0% FPR.
>
> Finally, regarding your point on our related work mentioning the persistance of watermarks, it appears that we need to clarify our point:\
> Watermarking is traditionally not made to radiate through the processes, only to hold information.
> While Sander et al. shows that watermarked text (via controlled sampling) can impact a model during fine-tuning, this behaviour is a fortunate byproduct of the initial goal: having detectable text. On the other hand, data taggants are hard to detect from clean data and their whole purpose is to impact models during training. This discussion is indeed interesting and we will make sure to clarify it.
>
> We sincerely hope that the reviewer can kindly consider _raising the score if our response helps address some of the concerns_.
>
> [1] Souri, Hossein, et al. "Sleeper agent: Scalable hidden trigger backdoors for neural networks trained from scratch." Advances in Neural Information Processing Systems 35 (2022)\
> [2] Geiping, J., Fowl, L., Huang, W. R., Czaja, W., Taylor, G., Moeller, M., & Goldstein, T. (2020). Witches' brew: Industrial scale data poisoning via gradient matching.\
> [3] Li, Y., Zhu, M., Yang, X., Jiang, Y., Wei, T., & Xia, S. T. (2023). Black-box dataset ownership verification via backdoor watermarking.

---

### Official Review · Reviewer_v3Ca · 2024-11-04

**Soundness:** 3
**Presentation:** 3
**Contribution:** 3
**Rating:** 6
**Confidence:** 4

**Summary:**

This paper introduces data taggants, a novel non-backdoor dataset ownership verification technique that helps detect if machine learning models were trained using a specific dataset. Unlike previous approaches that rely on backdoor watermarking, data taggants use pairs of out-of-distribution samples and random labels as secret keys, and employs clean-label targeted data poisoning to subtly alter a small portion (0.1%) of the dataset. When models are trained on the protected dataset, they respond to these key samples with corresponding key labels, allowing for statistical verification with only black-box access to the model. The authors validate their approach through comprehensive experiments on ImageNet1k using Vision Transformer and ResNet models, demonstrating that data taggants can reliably detect models trained on the protected dataset with high confidence, without compromising validation accuracy. The method proves to be stealthy, robust against various defense mechanisms, and effective across different model architectures and training recipes. It also provides stronger theoretical guarantees against false positives compared to previous approaches.

**Strengths:**

1. Use out-of-distribution samples as keys is quite novel.
2. Provides stronger statistical guarantees than previous work.
3. Well-structured methodology presentation.

**Weaknesses:**

1. Lacks formal security analysis against adaptive attacks.
2. No investigation of downstream task impacts

**Questions:**

1. How does the method defend against an adversary who knows the exact verification technique?
2. Why was 0.1% chosen as the modification budget, and how sensitive is the method to this choice?
3. Have you investigated potential negative effects on downstream tasks?

---

> ### Author Response · Authors · 2024-11-19
> **Response to Reviewer v3Ca**
>
> We thank the reviewer for their time and help.\
> We appreciate you found our work novel and recognise the stronger theoretical guarantees we provide compared to previous work.
>
> To address the above-mentioned weakness regarding a formal security analysis: could you please give precisions on what you would expect from such analysis and clarify what you call adaptive attacks?
>
> We are glad to address your questions:
> 1. We believe that by “the exact verification technique”, you mean the keys and the data taggants. If Bob:
>     - had knowledge of the keys, he could simply train on them with random labels;
>     - had knowledge of the data taggants, he could simply remove them from training.\
>   We remain at your disposal to include any other component you think could be considered as the exact verification technique.
> 2. Table 6 in appendix shows the performance of our method and baselines for different budget values (0.001%, 0.01%, 0.1%). The chosen budget of 0.1% corresponds to poisoning roughly 100 samples per key and is enough to be effective. Higher poisoning rates make the computation time too high to repeat them and run thorough experiments with standard deviations.
> 3. Given that we acknowledge tackling an image classification task with an image classification dataset, we fail to see what you would consider to be a “downstream task”. Could you please elaborate on this?
>
> We will be glad to address any remaining questions and concerns.

---

### Meta-Review · Area_Chair_TZdp · 2024-12-25

**Metareview:**

The submission  "Data Taggants: Dataset Ownership Verification Via Harmless Targeted Data Poisoning" proposes a dataset attribution via watermarking method using clean-label data poisoning. While reviewers point out that the exact algorithm used for clean-label poisoning is not new, this method is nevertheless an interesting application for the problem of data ownership that the authors examine carefully.

Based on this strength of the paper, I recommend acceptance.

**Additional Comments On Reviewer Discussion:**

The authors work with reviewers, such as 3ipm, through a number of concerns regarding the positioning of the work, and the writing regarding guarantees provided by these kinds of attribution methods. A few other, smaller concerns are resolved with reviewer CRSk.

The discussion with reviewer RFB3 brings up mainly the relationship to prior work in Guo et al. "Domain watermark: Effective and harmless dataset copyright protection is closed at hand." The discussion is interesting and I do think the papers are different enough. I expect that the authors extend their related work section with a more careful comparison.

For the record, I do not condone the tendentious AC messsage send to me to discredit RFB3, who is bringing up a valid concern, and was considering whether an ethics review was warranted. I do not think the tone of that message, and of the discussion with the reviewer, is necessarily a good one for this community, but I am judging this submission by the merit of its text.

---

### Decision · Program_Chairs · 2025-01-22

Accept (Poster)